health and disease and epidemiology, ecology

wildlife disease, *Toxoplasma gondii*, anthropogenic pressure, free-roaming cats, one health

**Author for correspondence:**
Amy G. Wilson
e-mail: amy.wilson@ubc.ca

# Human density is associated with the increased prevalence of a generalist zoonotic parasite in mammalian wildlife

Amy G. Wilson[1,2], Scott Wilson[1,3], Niloofar Alavi[4] and David R. Lapen[5]

[1]Department of Forest and Conservation Sciences, University of British Columbia, Vancouver, British Columbia, Canada V6T 1Z4
[2]Canadian Wildlife Health Cooperative, Abbotsford, British Columbia, Canada V3G 2M3
[3]Environment and Climate Change Canada, Delta, British Columbia, Canada V4 K 3N2 0H3
[4]Environment and Climate Change Canada, Ottawa, Ontario, Canada K1S 5B6
[5]Ottawa Research Development Centre, Agriculture and Agri-Food Canada, Ottawa, Ontario, Canada K1A 0C6

AGW, 0000-0003-2789-0480; SW, 0000-0002-1210-8727

Macroecological approaches can provide valuable insight into the epidemiology of globally distributed, multi-host pathogens. *Toxoplasma gondii* is a zoonotic protozoan that infects any warm-blooded animal, including humans, in almost every ecosystem worldwide. There is substantial geographical variation in *T. gondii* prevalence in wildlife populations and the mechanisms driving this variation are poorly understood. We implemented Bayesian phylogenetic mixed models to determine the association between species' ecology, phylogeny and climatic and anthropogenic factors on *T. gondii* prevalence. *Toxoplasma gondii* prevalence data were compiled for free-ranging wild mammal species from 202 published studies, encompassing 45 079 individuals from 54 taxonomic families and 238 species. We found that *T. gondii* prevalence was positively associated with human population density and warmer temperatures at the sampling location. Terrestrial species had a lower overall prevalence, but there were no consistent patterns between trophic level and prevalence. The relationship between human density and *T. gondii* prevalence is probably mediated by higher domestic cat abundance and landscape degradation leading to increased environmental oocyst contamination. Landscape restoration and limiting free-roaming in domestic cats could synergistically increase the resiliency of wildlife populations and reduce wildlife and human infection risks from one of the world's most common parasitic infections.

## 1. Background

Anthropogenic pressures can promote the emergence of novel pathogens or worsen the disease burden of endemic pathogens [1,2]. Examples of anthropogenic mechanisms creating these conditions include altering host community structure [1,3], reducing host health and immunity [4], disrupting ecosystem services capable of regulating pathogen transmission [5] and introducing invasive species and associated pathogens [6]. Human development and associated land-use changes can further amplify transmission risks by creating large, high-density populations of domestic animals that transmit pathogens to wildlife and humans across human–domestic animal–wildlife interfaces [1]. Although wildlife populations have increased pathogen richness, it is domesticated animals that have a more central role in sharing generalist pathogens with both humans and wildlife, ranging from viruses [7] to helminths [8] and ectoparasites [9]. Understanding how anthropogenic drivers influence the epidemiology of generalist pathogens is essential because they are more likely to be zoonotic, classified as emerging or reemerging diseases [10] and be of conservation concern [11].

*Toxoplasma gondii* is a model pathogen for examining how anthropogenic factors influence disease risk in wildlife populations across a range of taxonomic and geographical gradients. *Toxoplasma gondii* is a generalist protozoal parasite that is thought to be able to infect any endothermic animal, including humans, with a global human infection rate of 30–50% [12]. In addition to a broad host and geographical range, *T. gondii* has other key epidemiological traits such as using a domestic species as a definitive host, multiple transmission routes and extended persistence within hosts and the environment. *Toxoplasma gondii* infects hosts through fecal–oral exposure of a free-living, passively dispersed oocyst or carnivory of tissue cysts.

Domestic cats and wild felids serve as the definitive hosts in the domestic and wild cycles, respectively. Infected felids episodically excrete millions of environmentally robust oocysts in their faeces when they are exposed or re-exposed to infected prey or are immunosuppressed [13,14]. These oocysts can remain infectious for years in soil or water, passively dispersing through the soil matrix and contaminating aquatic ecosystems via surface runoff [15]. Infection occurs when animals or humans ingest oocysts in contaminated water, food or inadvertent surface contact. *Toxoplasma gondii* will permanently encyst within neurological and muscular tissue of infected hosts, with these tissue cysts being infectious to any carnivore that consumes that host. Transmission between intermediate hosts is a trait unique to *T. gondii*, enabling it to persist in the food chain, potentially circumventing what could have been a dead-end host [16].

*Toxoplasma gondii* infections can be fatal for immuncompromised humans or animals [17]. Multiple wildlife taxa are highly susceptible to *T. gondii* infections, such as marine mammals, Australian marsupials and New World monkeys, with reported mortalities in numerous endangered and endemic species [17]. However, even for healthy or insensitive hosts, latent *T. gondii* tissue cysts within host tissues can cause subclinical effects for the host's lifespan [17]. In humans, latent toxoplasmosis has associations with mental disorders (e.g. schizophrenia), epilepsy, autism, cognitive and vision deficits, cancers, traffic accidents and increased severity of other accompanying diseases [12].

Infected hosts can also have subtle behavioural changes affecting survival, such as reduced vigilance or even an attraction to felids, which for felid prey species would serve to complete the sexual life cycle of the parasite. These behavioural changes have been documented in species ranging from rats [18] to chimpanzees [19]. Although poorly understood, latent *T. gondii* infections in wildlife have been associated with reduced population fitness through altered fetal development [20] and increased mortality due to automobile collisions [21], cold weather [22] and concurrent parasitic infections [23]. Given these acute and chronic health impacts of *T. gondii* for such large numbers of individuals, identifying specific environmental risk factors that are amenable to mitigation, such as anthropogenic landscape changes and domestic animal management, would benefit public and wildlife health initiatives.

Due to the public and wildlife health burden of *T. gondii*, considerable research has been directed towards measuring prevalence in wildlife, with data now available for tens of thousands of individuals from hundreds of wild animal species worldwide. However, characterizing the probable drivers of these prevalence patterns has received less attention.

Since *T. gondii* prevalence data have been collected across a range of geographical gradients for diverse taxonomic groups, it is possible to integrate information from across studies to untangle specific climatic, anthropogenic, ecological and phylogenetic correlations with *T. gondii* prevalence. Moreover, the broad host range, coupled with global data richness, enables a global perspective that is rarely possible for other individual wildlife pathogens.

In this paper, we test three sets of hypotheses related to the ecological, climatic and anthropogenic correlates of *T. gondii* prevalence in mammalian wildlife. We first tested the hypothesis that species-specific ecological traits related to ecosystem and trophic level would lead to different exposure rates to *T. gondii* through oocysts and tissue cysts. Species living in aquatic ecosystems are expected to have increased oocyst exposure risks due to the potential for substantial and localized oocyst influxes through runoff and increased exposure area through suspension through the water column. Aquatic avian wildlife has an increased prevalence of *T. gondii* relative to terrestrial species [24], and we predicted this pattern would also be evident at a global scale in mammals. Trophic level influences exposure risk because carnivores of endothermic vertebrates will be exposed to increased infection risks through the tissue-cyst transmission route. Similarly, studies have reported a consistent positive relationship between *T. gondii* prevalence and trophic levels [24,25]; therefore, we predicted that prevalence would be higher among carnivorous taxa.

Our second set of hypotheses considered climatic factors since temperature and precipitation affect oocyst survival and transport in the environment and domestic cat abundance and free-roaming activity [26]. Extreme temperatures can reduce oocyst survival [27], while heavy precipitation facilitates oocyst transport in terrestrial and aquatic ecosystems [28]. Therefore, we predicted a positive association between *T. gondii* prevalence in mammals and the average air temperature and precipitation at the locations where they were surveyed.

Finally, our last and most important hypotheses examined the relationship between *T. gondii* prevalence and anthropogenic factors. Mammals in more anthropized locations may have a higher *T. gondii* prevalence because of land conversion (e.g. impervious surfaces and losses of wetlands [5,29]) effects on oocyst transport and survival and increased exposure to higher densities of free-roaming domestic cats [30], which are the most consequential definitive host species compared to wild felids [28]. To measure the relationship between *T. gondii* and prevalence in mammals in anthropogenic environments, we selected three spatial layers (i) human density representing urbanization, (ii) crop coverage representing agricultural environments and (iii) the human footprint index (HFI) as a composite measure of multiple human pressures. We predicted a positive relationship between *T. gondii* prevalence and these three measures of human impact.

## 2. Methods

### (a) Literature search

Our analyses focused on studies reporting on the prevalence of *T. gondii* in free-ranging wild mammal populations. We searched for studies using Web of Science Core Collection and PubMed using search terms: 'wild\*' and 'toxoplasmosis' or 'toxoplasma.'

We located additional studies from the reference lists of these studies, citations of these works and comprehensive review articles [17]. As we were interested in the geographical variables at the sampling site, we only included studies that provided specific location information or where the sampling locations could be estimated within a 50 km distance. For our analyses, we selected a 50 km threshold in order to reflect the average precision of sampling information provided by authors and to more realistically capture the geographical gradients experienced by individuals. Of the 238 species in this study, a subset of 142 had species-specific home range estimates available [31]. The home range estimates were much smaller than 50 km for 127 of these species, with individuals from these taxa constituting 78% of the full dataset and 93% of the subset of taxa with estimated home ranges (electronic supplementary material, table S1). We excluded pelagic marine mammals due to their propensity for long-distance movements, which introduces uncertainty regarding the geographical characteristics of their likely exposure location. We also excluded studies that reported only seropositive cases, provided only genus level identifiers or were from captive or farmed wild animals. The complete list of included studies is provided in electronic supplementary material, table S2, and the study inclusion path is provided in electronic supplementary material, figure S1.

From each study, we extracted data on the number of positive and negative animals, the total number of individuals tested, the species tested, method of detection (serology or isolation) and geographical information of the sampling location. For each species, we then compiled species-specific ecological traits. To account for ecological correlates of T. gondii prevalence in mammals, we collected information on species ecosystem type (terrestrial or aquatic) from the IUCN Red List [32]. The trophic level of a species was based on the proportion of a species's diet consisting of fruit, nectar, seeds, invertebrates and vertebrate prey as provided in EltonTraits 1.0 [33]. We reclassified these proportions into four categories: herbivores, invertivores, omnivores and carnivores. Herbivores and invertivores were species where greater than 50% of the diet was vegetation and invertebrate prey, respectively. We defined omnivores as species where the primary diet could be vegetation or invertebrates but included less than 50% vertebrates. Carnivores were species that primarily consumed greater than 50% live vertebrate prey or carrion. Species-specific dietary compositions and resulting classifications are provided in electronic supplementary material, table S1.

For all sampling groups within each study, we extracted location-specific climatic data from the WorldClim 2.0 database [34], human population density data from the Center for International Earth Science Information Network (CIESIN) [35], land coverage from the European Space Agency Climate Change Initiative Land Cover maps (CCILC) [36] and a composite measure of anthropogenic pressure from the HFI database [37]. For the WorldClim database, we obtained spatial data for interpolated estimates of average annual temperature and precipitation (1970–2000) at a 10 km resolution. The average, maximum and minimum temperatures were highly correlated, and exploratory analyses showed that average temperature was a stronger predictor of T. gondii prevalence than the maximum or minimum temperature; therefore, we retained average temperature for all analyses.

We obtained spatial data on human population density (people per $km^2$) from the CIESIN, using the 1 km resolution maps. Land cover data were obtained from the CCILC database with a 300 m resolution. For both the human density and land cover maps, we obtained data for 2000, 2005, 2010, 2015 and 2019 and for each study, we used the time period for those variables that most closely corresponded to the sampling dates of the study. Site-specific cropland coverage was calculated from the 22 land cover classes provided in the ESA CCILC database, where

we included the four cropland cover classes of rainfed cropland, irrigated or post-flooding cropland, mosaic cropland and mosaic natural vegetation. A composite measure of anthropogenic pressures was obtained from the 2009 Human Footprint map, which is a 1 km resolution map of the cumulative anthropogenic pressures arising from human density, built environments, electric infrastructure, croplands, pasture lands, transport routes (road, railways and navigable waterways) [37] termed as the HFI.

We extracted the average annual temperature and precipitation, human density and HFI for each sampling location averaged over a 50 km buffer around the sampling point using the R package raster [38], with a World Geodetic System 1984 projection. Using ArcGIS [39], cropland was calculated as the proportion of the 50 km buffer area that contained any of the four CCILC cropland types. The buffers were transformed to the local UTM coordinate system of each site to minimize area distortion, especially for the sites in the higher latitudes. A 25 km buffer was also calculated to determine comparability. All geographical fixed-effect variables were standardized before analysis to place them on the same scale, and we calculated Pearson correlation coefficients between standardized fixed variables prior to testing (electronic supplementary material, table S3). There were no variables appearing in the same model with a mean correlation above 0.7. For temperature and cropland, which had a correlation of 0.69, we compared all model coefficients when the two variables were included together and separate. There were only small differences in the coefficients for the two scenarios, and therefore we did not exclude these correlated variables from appearing in the same model. The sampling representation across the climatic and anthropogenic variables is provided in electronic supplementary material, figure S2.

## (b) Statistical analysis

We tested our hypotheses using Bayesian phylogenetic generalized linear mixed models as implemented in the R package MCMCglmm [40], following the methods of Barrow et al. [72]. These models enable the incorporation of phylogenetic distance to accommodate for non-independence due to shared ancestry. Phylogenetic variance was included as a random effect with the phylogenetic covariance matrix for mammalian taxa based on 1000 birth–death node-dated trees [41], with the R package ape [42]. Prevalence data were modelled as counts of positive individuals and negative individuals with an assumed multinomial distribution. We took a hypothesis testing approach and evaluated the relative support for an association between T. gondii prevalence and ecological, climatic and anthropogenic factors. Because our expectations were for linear relationships between the predictor variables and prevalence, we did not test curvilinear relationships, nor did we include interactions among the predictor variables. Model support was evaluated based on the deviance information criterion (DIC), where a difference in DIC (ΔDIC) of greater than 3–7 would be indicative of less model support [43]. Fixed-effect predictors were considered to be significant if the 95% credible interval (CI) did not overlap with zero.

Model selection started with an intercept-only model that was used to evaluate support for a mixed-effects base model with study, species and phylogeny as random effects and detection method (serology versus isolation) as a fixed effect. This mixed-effects base model had substantial support over the intercept-only model (ΔDIC = 366.35) and was therefore included in all subsequent model testing. The analyses to test our three main hypotheses first assessed an ecological traits-only model, evaluating the support for an association of ecosystem or diet, separately or combined on T. gondii prevalence. Similarly, a climate-only model was tested, where we examined the relative relationship between T. gondii prevalence and annual average precipitation and temperature. For both of these model comparisons, the

**Table 1.** Deviance information criterion (DIC) model selection results for testing hypotheses for ecological (eco and habitat), climatic (temp and precip) and anthropogenic (human, crop and HFI) risk factors associated with *Toxoplasma gondii* prevalence patterns in free-ranging mammalian wildlife. ΔDIC = change in DIC relative to the top model in each hypothesis. Predictor fixed effects: eco = aquatic or terrestrial; diet = herbivore, invertivore, omnivore or carnivore; temp = average annual temperature; precip = average annual precipitation; HFI = human footprint; crop = cropland coverage; human = human density. The base model (*base*) includes detection method as a fixed effect and random effects for phylogeny, species and study (DIC = 30 584.59). See Methods for further details.

| | | DIC | ΔDIC |
|---|---|---|---|
| **1. ecological variables—ecosystem and diet** | | | |
| ecosystem only | eco + *base* | 30 583.9 | 0 |
| ecosystem and diet only | eco + diet + *base* | 30 586.21 | 2.31 |
| **2. climatic variables—precipitation and average temperature** | | | |
| temperature only | temp + *base* | 30 576.73 | 0 |
| temperature and precipitation | temp + precip + *base* | 30 618.58 | 41.85 |
| **3. anthropogenic variables—human density, HFI, cropland coverage** | | | |
| human density | eco + temp + human + *base* | 30 552.19 | 0 |
| HFI | eco + temp + HFI + *base* | 30 569.93 | 17.73 |
| ecosystem and temperature | eco + temp + *base* | 30 576.17 | 23.97 |
| cropland | eco + temp + crop + *base* | 30 576.94 | 24.75 |

statistically significant predictors were then retained in models testing the relative correlation of *T. gondii* prevalence with anthropogenic variables. Using the mixed-effects base model and including influential ecological and climatic variables, we compared model support and mean effects for human population density, crop coverage and HFI (table 1). The top selected model was used to estimate mean effects of all fixed-effect variables and the relative contribution of phylogenetic and non-phylogenetic species' effects to the total random variation, which approximates the phylogenetic signal (termed lambda, $\lambda$).

All models assumed a multinomial distribution and were run for 400 000 iterations with a burn-in of 100 000 and a thinning interval of 100. For the random effects, we used the priors as implemented in MCMCglmm ($V = 1$ and $v = 0.02$). All variables had effective sample sizes exceeding 2500, Gelman statistics approximating 1.0 and consistent trace plots, all of which indicate MCMC convergence [44]. The R package *epiR* [45] was used to calculate *T. gondii* prevalence at the taxonomic family level (electronic supplementary material, table S4).

## 3. Results

Our literature search produced 202 *T. gondii* prevalence studies representing 238 species from 54 taxonomic families and 45 079 tested individuals. Collectively, these studies sampled individuals from 981 locations across the globe (figure 1). There were notable study gaps in Asia (i.e. Russia, China, India), but similar absolute latitudinal gradients were captured in North and South America and from northern Scandinavia to equatorial Africa. The number of sampled individuals from different dietary-ecosystem groups were as follows: aquatic herbivore ($n = 1433$), aquatic omnivore ($n = 25$), aquatic carnivore ($n = 1443$), terrestrial herbivore ($n = 24\,678$), terrestrial invertivore ($n = 1522$), terrestrial omnivore ($n = 13\,423$), terrestrial carnivore ($n = 2555$).

Compared to an intercept-only model, the addition of study, species and phylogeny as random effects led to significant model improvement ($\Delta$DIC = 366.35). For the ecological trait-only (ecosystem and diet) and climatic factor-only (temperature and precipitation) models, only ecosystem type and

temperature were significant, both improving model support over the base model with their inclusion ($\Delta$DIC = 8.42). Among the anthropogenic variables, the inclusion of human density resulted in the greatest improvement in model support ($\Delta$DIC = 23.97), followed by the inclusion of HFI ($\Delta$DIC = 17.73). By contrast, the inclusion of crop coverage did not improve model support over a reduced model (table 1). The top model containing human density, temperature and ecosystem (table 1), showed a significant and positive effect of human density ($\beta = 0.71$; 95% CI: 0.41–1.03) and average temperature ($\beta = 0.43$; 95% CI: 0.23–0.63) on *T. gondii* prevalence. Mammalian species in terrestrial ecosystems had a lower average prevalence than species in aquatic ecosystems ($\beta = -0.96$; 95% CI: −1.69 to −0.25; figure 2). Results for model selection and mean effect coefficients were comparable if a 25 km buffer was used (electronic supplementary material, table S5). We found a significant phylogenetic signal for *T. gondii* prevalence, where phylogenetic variation ($\lambda$) accounted for 48% of the random variation (95% CI: 0.25–0.68), while the variation attributed to species as a random variable accounted for approximately 3.9% of total variation (95% CI: 0.00–0.14).

## 4. Discussion

Our global analysis supported our hypotheses that *T. gondii* prevalence would be associated with human density, temperature and ecosystem type across a diversity of mammalian taxa and a broad geographical range. The most compelling result was strong support for our hypothesis that *T. gondii* infections in wildlife would be greater in areas of higher human density. This finding is consistent with previous finer geographical-scale studies that have documented increases in *T. gondii* prevalence in wildlife with increasing anthropogenic pressures [46–49]. Increased human density (rural and urban development) is connected with multiple landscape alterations that could influence the epidemiology of *T. gondii*, the most intuitive being the abundance of owned and unowned free-roaming domestic cats [50]. Although such a direct association between cat density and *T. gondii* prevalence cannot be made at

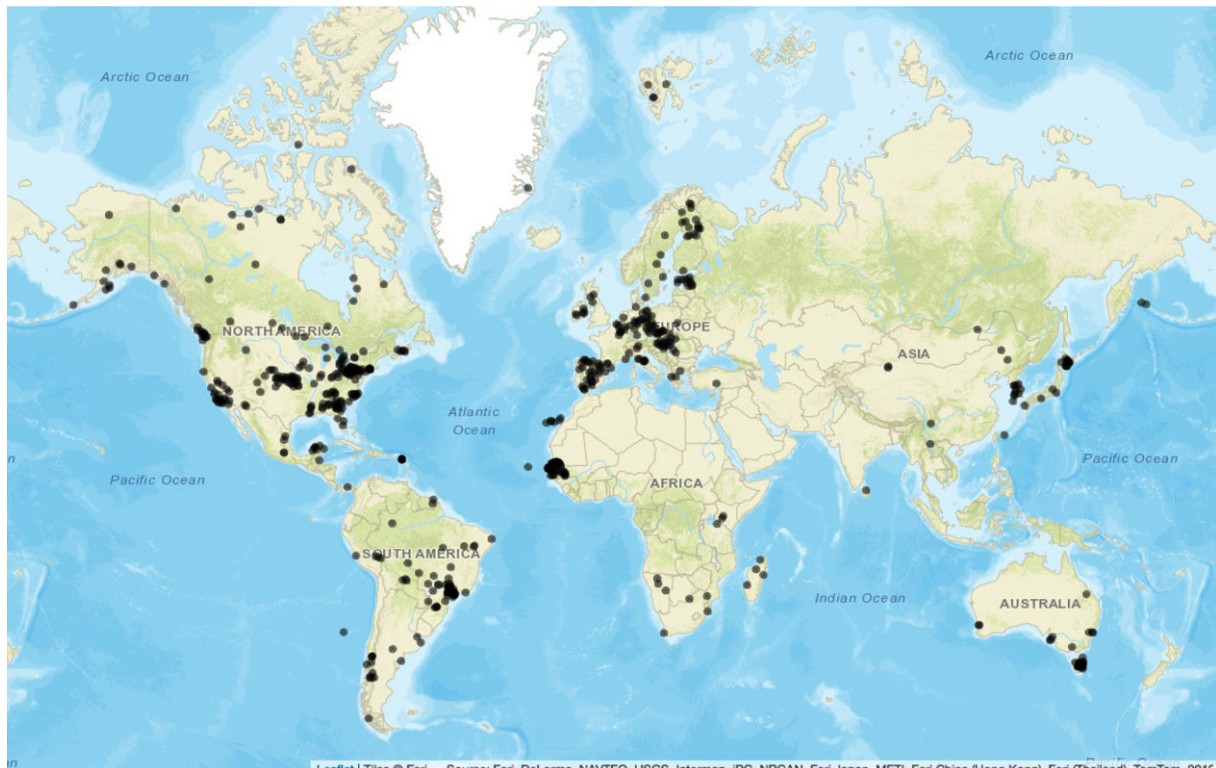

**Figure 1.** Distribution of study sites included in the global analysis of *Toxoplasma gondii* prevalence data for free-ranging wild mammal populations. (Online version in colour.)

the global scale due to a lack of cat abundance data, this association has been made at local scales [21,51,52].

There are an estimated 600 million domestic cats globally [53], which is several orders of magnitude larger than the combined abundance of all wild felid species [32]. Consequently, although multiple species of felids are known definitive hosts for *T. gondii*, this significant disparity in population size, and the positive association between *T. gondii* prevalence and human density in this present study, provide additional support for domestic cats being the most consequential definitive host for wildlife *T. gondii* infections [28]. Wildlife populations in pristine environments with exposure only to wild felids have a lower *T. gondii* prevalence than populations with increased exposure to domestic cats [21,52]. Therefore, *T. gondii* transfer from domestic cats into wildlife populations likely elevates infection rates above more naturalized endemic levels and introduces domestic strains into the wildlife cycle. The transfer of *T. gondii* strains between wildlife and domestic cats is consequential because strains predominating in domestic cats versus wild felids differ in virulence, shedding behaviour and oocyst and tissue-cyst infectivity [16]. Compared to wild-type strains, domestic cat strains have evolved increased tissue-cyst infectivity [16] and induce more prolific oocyst shedding in domestic cats [54]. These traits of domestic strains could further increase the prevalence and persistence of *T. gondii* in exposed wildlife populations.

Increases in human density also impact the severity and spatial scale of *T. gondii* contamination by removing ecosystem services that reduce oocyst spread and limit the size of free-roaming cat populations. For example, wetlands and other naturalized features can help impede lateral oocyst spread via terrestrially based runoff into aquatic systems [5], functions which are crippled in landscapes dominated by impervious surfaces [29]. Intact ecosystems are also

associated with higher abundances of native predators such as coyotes that limit the infiltration of free-roaming cats into important wildlife areas [55]. This protective role of native predators was demonstrated when population declines of Tasmanian devils (*Sarcophilus harrisii*) were correlated with both an increase in the numbers of feral cats and *T. gondii* prevalence in native wildlife [21].

Due to the significant landscape and hydrological alterations associated with agricultural intensification, we had predicted that *T. gondii* prevalence in wildlife would be positively associated with cropland cover. However, we did not find a consistent association with cropland coverage. Native habitat retention within agricultural landscapes has been associated with lower risks of pathogen transfer from domestic animals into wildlife [56]. However, domestic cat densities frequently show a reduced [57] but unequal distribution across rural habitat types [50], and this may complicate attempts to elucidate a broad-scale pattern using available land cover metrics.

We also found support that *T. gondii* prevalence is influenced by climate. Specifically, we found that *T. gondii* prevalence increased in warmer locations. Temperature may influence prevalence through both variation in oocyst survival and host distribution [58]. Oocyst viability declines at temperatures below −20°C [27], but colder regions could also have smaller free-roaming domestic cat populations due to decreased overwinter survival of feral cats and owned cats being limited to free-roaming only during warmer seasons. Although precipitation has been demonstrated to increase *T. gondii* transmission to the aquatic ecosystem [59,60], we had anticipated increased precipitation would also be associated with increased exposure within terrestrial ecosystems. Contrary to our predictions, precipitation lacked a significant association with *T. gondii* prevalence. The role of precipitation

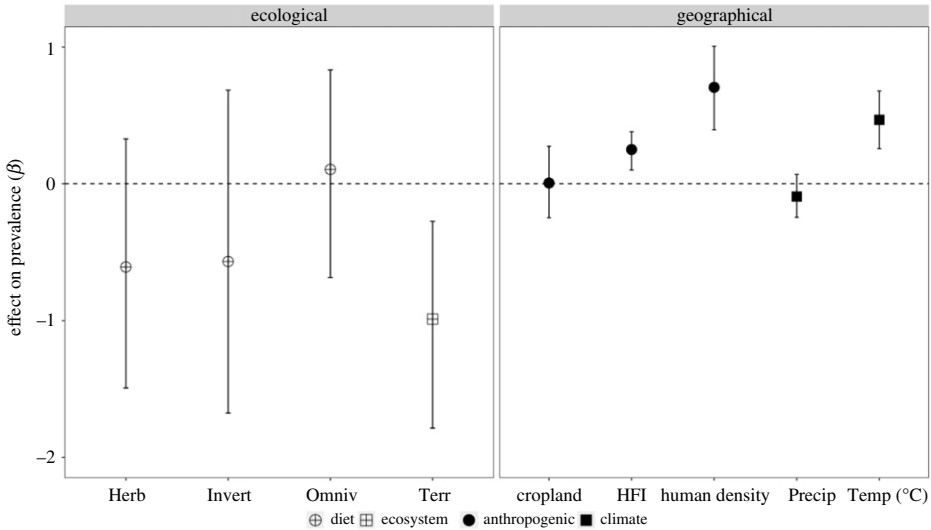

**Figure 2.** Posterior mean effects (95% credible intervals shown) for fixed variables in a top model containing the ecological, climatic and anthropogenic factors tested for an association with *Toxoplasma gondii* prevalence in global free-ranging wild mammal populations. The ecological species-specific variables shown in the left facet are the effect on prevalence for mammals that have primarily a herbivore (Herb), invertivore (Invert) and omnivore (Omniv) diet relative to a carnivore diet and live in a terrestrial (Terr) relative to an aquatic ecosystem. The geographical variables in the right facet are cropland coverage, human footprint (HFI), human population density, annual precipitation (Precip) and average annual temperature ℃ (Temp).

in the fate and transport of oocysts would be impacted by local hydrological governing factors, such as soil and topography, dilution effects, seasonality and the nature and degree of precipitation intensity and antecedent soil moisture conditions [59,60].

Waterborne toxoplasmosis via oocysts is an important route of infection for humans and wildlife [15,24], and we did find that aquatic mammals had an increased *T. gondii* prevalence compared to terrestrial species. Because there is no known aquatic definitive host for *T. gondii*, aquatic animals are likely exposed to oocysts transferred from felids in terrestrial ecosystems [28]. Studies of California sea otters (*Enhydra lutris*) [60] and beluga whales (*Delphinapterus leucas*) [61] reported increased *T. gondii* prevalence under conditions of higher terrestrial runoff. Further evidence of anthropogenic oocyst loading of freshwater comes from the higher prevalence of *T. gondii* in muskrats (*Ondatra zibethicus*) exposed to wastewater [62] or watersheds subject to larger amounts of terrestrial runoff [63], compared to the absence of seropositive muskrats in pristine watersheds [64]. Compared to terrestrial ecosystems, there may also be increased oocyst accumulation and persistence within aquatic ecosystems due to oocyst adherence to biofilms on aquatic vegetation [65] and bioaccumulation within aquatic microorganisms [66] and larger-bodied filter-feeding transport hosts [67]. This bioaccumulation of waterborne pathogens has been leveraged in novel surveillance programmes for both marine [68] and freshwater ecosystems [69].

By contrast to comparative studies of sympatric taxa [25], we did not find a significant effect of a species' trophic level, suggesting that environmental exposure (oocyst-associated) may be more globally significant than trophic transmission (tissue-cyst-associated). Tissue cysts could have an inherently lower transmission rate because infections require that a predator ingest viable tissue cysts, which may infrequently occur unless the entire body of the prey is freshly consumed. Results from this study and other large-scale analyses [4] highlight how geographical factors such as anthropogenic pressures in an individual's ecosystem may be more

predictive of disease risk than generalized species-specific ecological traits. For example, within this dataset, black bears (*Ursus americanus*) had double the prevalence (58%) of the less synanthropic, but similarly omnivorous, grizzly bears (*U. arctos*, 24%). Similarly, insectivorous bats (*Miniopterus schreibersii*), which would be expected to be a low-risk ecological guild for *T. gondii* exposure, had a significantly higher prevalence in southern Myanmar compared to conspecific populations in China, again demonstrating the strong influence of geographical factors [70,71].

Despite the variable impact of species ecology on *T. gondii* prevalence, we detected a notable phylogenetic signal. Although few studies have quantified a phylogenetic signal for wildlife diseases, which limits comparison, the signal for *T. gondii* was higher than that reported for a smaller scale study focused on avian malaria among multiple orders of Andean birds (λ: 0.13–0.35 [72]). The phylogenetic signal for a generalist pathogen such as *T. gondii* probably reflects evolutionarily conserved ecological or physiological host traits that influence prevalence but were not included in our analysis. For example, the evolutionarily distant families, Didelphidae and Procyonidae, are both synanthropic, terrestrial omnivores, but the former has half the mean prevalence, suggesting the presence of other unaccounted for drivers. Additionally, the clinical severity of *T. gondii* infections varies among taxonomic groups, with related taxa showing similar patterns of susceptibility, which would also contribute towards a phylogenetic signal in this dataset [72].

We acknowledge several important caveats in our study. First, as we noted earlier, although a large amount of *T. gondii* prevalence data were available, there are key areas such as central Eurasia and east-central Africa that had minimal representation, which is unfortunate given that countries on these continents have relatively high human *T. gondii* prevalence [12]. Furthermore, we broadly categorized ecosystem types as terrestrial or aquatic, and sampling efforts are unlikely to be uniform across different environments and habitats within these ecosystems. Therefore, the associations

found in the present analysis may differ from those present in wildlife populations from data-deficient regions or environments. An emphasis on georeferencing data and passive surveillance of wildlife populations for underrepresented taxa, regions and ecosystems would enable a more refined analysis for additional ecological and geographical variables that may be associated with *T. gondii* prevalence.

Second, while our model can help identify prospective higher risk areas for *T. gondii* associated with climate and human densities, we acknowledge that the identification of cause and effect mechanisms, such as domestic cat abundance, can only be deduced in a general manner at a global spatial scale. Molecular epidemiological methods are well suited to address these knowledge gaps with increasingly sensitive assays for detecting *T. gondii* DNA in environmental samples [15]. Moreover, these studies can genotype *T. gondii* isolates and identify them as strains most likely to originate from wild or domestic cycles [73]. Even in the absence of mechanistic cause and effect evidence herein, the presence of a significant relationship between human density and *T. gondii* prevalence in wildlife at a global scale suggests that proactively targeting pathogen pollution from domestic cats would be the most pragmatic and impactful intervention for decreasing wildlife infections. Approaches targeting intermediate hosts fail to address the significant role of oocyst-associated infections in *T. gondii* epidemiology. For example, focusing management efforts on removing pathogen inputs from the domestic pig definitive host led to a significant reduction in *Trichinella spiralis* prevalence in European wildlife in less than a decade [74].

Adopting an ecosystem-based approach for managing zoonotic diseases can be more economically efficient than reactionary interventions to single pathogens [75], with synergistic health and conservation benefits. Addressing ecosystem health may be particularly efficacious for *T. gondii* prevention. Free-roaming domestic cats depredate tens of billions of wild animals each year [76], with additional wildlife and public health impacts through the transmission of *T. gondii* and other pathogens [77]. Mitigating these influences through landscape restoration and effective population management of free-roaming cats would simultaneously benefit wildlife through reduced predation mortality, disease and increased population resiliency. Furthermore, because wildlife are reliable sentinels for human exposure risk to *T. gondii*, these actions could also contribute towards decreasing the human health burden of one of the most common global human parasitic zoonoses.

Data accessibility. Master dataset used for analysis have been included as electronic supplementary material. The data are provided in the electronic supplementary material [78].

Authors' contributions. A.G.W.: conceptualization, data curation, formal analysis, investigation, methodology, validation, visualization, writing—original draft, writing—review and editing; S.W.: conceptualization, formal analysis, investigation, writing—original draft, writing—review and editing; N.A.: formal analysis; D.R.L.: conceptualization, formal analysis, funding acquisition, investigation, writing—original draft, writing—review and editing. All authors gave final approval for publication and agreed to be held accountable for the work performed therein.

Competing interests. We declare we have no competing interests.

Funding. This work was financially supported by Agriculture and Agri-Food Canada Project J-002305, Environmental Change One Health Observatory (ECO$^2$)

Acknowledgements. We gratefully acknowledge the efforts of all authors whose published work form the basis of our global analysis. We also thank the anonymous reviewers whose comments and suggestions greatly improved this manuscript.

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
