## [Peer Review File · Proceedings of the Royal Society B: Biological Sciences]

Review History

RSPB-2020-2551.R0 (Original submission)

Review form: Reviewer 1

Recommendation

Accept with minor revision (please list in comments)

Scientific importance: Is the manuscript an original and important contribution to its field?

Good

General interest: Is the paper of sufficient general interest?

Good

Quality of the paper: Is the overall quality of the paper suitable?

Good

Is the length of the paper justified?

Yes

Should the paper be seen by a specialist statistical reviewer?

No

Do you have any concerns about statistical analyses in this paper? If so, please specify them explicitly in your report.

Yes

It is a condition of publication that authors make their supporting data, code and materials available - either as supplementary material or hosted in an external repository. Please rate, if applicable, the supporting data on the following criteria.

Is it accessible?

Yes

Is it clear?

No

Is it adequate?

Yes

Do you have any ethical concerns with this paper?

No

Comments to the Author

The study "Human footprint influences the global prevalence of a generalist parasite in mammalian wildlife" aims to explore variation in the infection of mammalian wildlife species with *Toxoplasma gondii* at global scale, compiling evidence from published field surveillance data and using a regression analysis, with host species traits, climate variables and a human footprint index as covariates.

I found this study a timely piece of work and a very well prepared manuscript. A possible shortfall, however, is in my opinion that the authors did not account for any possible spatial effect and sampling bias in their analysis. According to Figure 1, the underlying data are mostly clustered in N America and Europe, and I therefore think that such possible biases should be explored..

Perhaps the authors could also have put more efforts into distinguishing the different wildlife species in terms of their habitat use in addition to using an index of human food print in their analysis, as many wildlife species that are frequently recorded in anthropogenic habitats are known to share more pathogens with domestic animals and humans; this shortfall should at least be discussed if not included in an amended analysis.

There are multiple rows in the provided data table (file "MS_Data") with both zero for "pos" and "neg", which was unclear to me and needs clarification. Are the metadata for the data accessible?

Additional comments:

Line 37: I do not understand "ecological restoration" in context of your argument.

Line 55 and elsewhere: Given the importance of domestic cats in *Toxoplasma* transmission cycles, you should consider to extent your argument to the human-domestic animal-wildlife interface, in line with the growing number of studies exploring the importance of domestic and commensal animals for host sharing and pathogen spillover. Some recent study to incorporate on evidence of domestic species and anthropogenic habitat in pathogen host shifting and spillover;

Gibb, R., Redding, D.W., Chin, K.Q., Donnelly, C.A., Blackburn, T.M., Newbold, T. & Jones, K.E. (2020) Zoonotic host diversity increases in human-dominated ecosystems. *Nature*, 584, 398-402.

Wells, K., Morand, S., Wardeh, m. & Baylis, M. (2020) Distinct spread of DNA and RNA viruses among mammals amid prominent role of domestic species. *Global Ecology and Biogeography*, 29, 470-481.

Line 160: Its was unclear to me how you matched the terrestrial human footprint index to marine species such as *Beluga* or *Dugongidae*?

Line 201: Table 1 could be presented as supplementary material?

Lines 202-203: Did you test models with all different combination of variables? Your statement here to compare an intercept only model with one including ecological trait and climate variables leaves the question of how a model with human footprint HPF but not ecological trait and

climate variable would compare to other models?

Lines 204-201: What do these posterior effects mean? Can they be expressed as odds ratio or % change in prevalence?

Lines 215-220: Perhaps worth to explore this phylogenetic effect further and mention any taxonomic groups with particularly high/low prevalence records?

Line 336: Please check if Dobson 2020 has been included in reference list.

Review form: Reviewer 2

Recommendation

Major revision is needed (please make suggestions in comments)

Scientific importance: Is the manuscript an original and important contribution to its field?

Acceptable

General interest: Is the paper of sufficient general interest?

Good

Quality of the paper: Is the overall quality of the paper suitable?

Good

Is the length of the paper justified?

Yes

Should the paper be seen by a specialist statistical reviewer?

Yes

Do you have any concerns about statistical analyses in this paper? If so, please specify them explicitly in your report.

No

It is a condition of publication that authors make their supporting data, code and materials available - either as supplementary material or hosted in an external repository. Please rate, if applicable, the supporting data on the following criteria.

Is it accessible?

Yes

Is it clear?

No

Is it adequate?

No

Do you have any ethical concerns with this paper?

No

Comments to the Author

General Comments

This is an interesting study in that it seeks to analyze factors related to *T. gondii* presence at a global scale. The power of the work is in the size of the database being analyzed. However, while you have a large database, it was somewhat unclear why this work is as important? In essence, the Introduction did not do a good job of indicating why the work you pursued was really

needed beyond what we already know, especially if looking for factors that explain relationships. Aside from the lack of set up, my main issue with the work is that it presents itself as mechanistic in a way that is beyond the data. I know there is a divide between how spillover has been presented outside of ecology versus within the discipline, but the variables analyzed in this study really can only demonstrate if there are correlational relationships, not what are driving factors. As a result, there are not only a lot of caveats, but a number of definitive statements cannot be made. In addition, the lack of data on cats for your study sites mean that we cannot accurately test the relationship between HFI and *T. gondii*. All this is to say that I see value in the research, but that it is need of much better contextualization and stating what piece of knowledge it is adding to that we are lacking.

Specific Comments

L51. Suggest citing Chalkowski, K., C.A. Lepczyk, and S. Zohdy. 2018. Parasite ecology of invasive species: new directions. *Trends in Parasitology* 34:655–663.

L58. It is really unclear how this is related to *T. gondii* as haven't really described the linkages here.

L62. Provide citations here to support.

L62. Ok, but aside from mortality is behavior change. In both humans and wildlife there is quite a bit of data on how *T. gondii* changes animal behavior and this is something of great importance. I would suggest that you describe more about the parasite in terms of health implications as well. It isn't just mortality we worry about in humans.

L67. A big issue with wildlife is its impact on threatened and endangered species or species that are endemic. For instance, *T. gondii* is a large threat to many island animals.

L68. I don't see the connection to roads as written.

L73. Be specific and name house cats as the definitive host.

L86. Incomplete paragraph, need at least three sentences here.

L99. Anthropogenic?

L101. I wouldn't quite call a lot of this spillover. Also, consider how oocysts move in the environment, they can easily move in streams and across land in areas with rain and topography. For instance, see work by Van Wormer.

L104. This paragraph doesn't really connect to previous one. Also, I didn't see a set up prior to this paragraph as what the larger need is for your study. That is, what makes this work novel or needed?

L114. And temperature influence the number, survival, and prevalence of outdoor cats that have a large input of oocysts into the environment.

L128. Which form of Web of Science did you use as type matters. See Calver MC, Goldman B, Hutchings PA, Kingsford RT. 2017. Why discrepancies in searching the conservation biology literature matter. *Biological Conservation* 213:19-26.

L129. Why not also use '*T. gondii*' variations?

L148. Need to describe how Elton Traits database categorizes trophic level numerically (what are values and what do they mean) so that reader can see link to your prediction.

L159. Why averages and not also minimums and maximums? Considering that both temp and precip factor into *T. gondii* distributions, wouldn't it have been important to evaluate these other measures as well, especially if trying to find which factor may be more important?

L160. Why use HFI over other types of measures, such as land cover, nighttime lights, pop density? I have some idea, but I also find that HFI can have some problems with it as you are lumping a lot of ideas into a single index. Need to justify.

L164. Buffer around what? Why 50 km? Also, need to describe the pixel size or minimum mapping unit of your GIS data as well as what datum and projections are being used.

L202. What are ecology variables?

L211. Move to Discussion.

L224. Please begin Discussion with answers to your predictions.

L226. I would disagree with this conclusion. Your work is entirely corollary and not mechanistic. A large missing component of your research is the number or density of cats in the environment or on the landscape. Essentially you are equating HFI with cat numbers, which may not always be the case. Thus, you may have found support for a prediction, but you have not really

identified the mechanism here. I would certainly not call it anthropogenic spillover.

L243. This paragraph is very speculative. Some of these land cover types would also simply not have cats, so is it conversion versus fact that cats are less abundant in wetlands? In the end this paragraph comes across as more of a post hoc rationalization of the analyses as you did not look at land cover/use change or other variables discussed here. This is the tradeoff of using HFI.

L294. But how did those studies measure trophic levels? One problem with trait data is that different researchers use different values and not all of these studies may be comparable.

L321. Again, I would disagree with this point. You essentially have a large database that is more akin to biogeography research in which you can test patterns along a descriptive nature that support or refute hypotheses. But you cannot determine mechanisms here. This is a large misunderstanding of quite a bit of the spillover literature in terms of how landscape ecology, biogeography, and macroecology evaluate the relationships.

L328. Agree with all of this, but wasn't really the point of the ms.

Table 1. Did you begin with an explicit framework of a priori models to consider that were based on your predictions? If so, please list these models somewhere.

Figure 1. One item that is apparent from this figure is that there may be some notable bias in locations sampled. There are few areas that have a low HFI and also large numbers of biomes missing.

Decision letter (RSPB-2020-2551.R0)

23-Nov-2020

Dear Dr Wilson:

I am writing to inform you that your manuscript RSPB-2020-2551 entitled "Human footprint influences the global prevalence of a generalist parasite in mammalian wildlife." has, in its current form, been rejected for publication in Proceedings B.

This action has been taken on the advice of referees, who have recommended that substantial revisions are necessary. With this in mind we would be happy to consider a resubmission, provided the comments of the referees are fully addressed. However please note that this is not a provisional acceptance.

Sincerely,
 Professor Hans Heesterbeek
 mailto: proceedingsb@royalsociety.org

Associate Editor

Board Member: 1

Comments to Author:

Based on two expert reviews and my own evaluation of the manuscript, there is a consensus that there is much value in the manuscript submitted by Wilson et al., but a number of issues have been raised that require an extensive revision of the manuscript:

- 1) The supporting data, while accessible, are difficult to interpret and require clarification.
- 2) Reviewers raised concerns about whether the Human Footprint Index is a good measure of domestic cat abundance and this comment should be addressed in detail.
- 3) Additional details on the factors included in the analysis and other analytical decisions require further justification.
- 4) The framing of the manuscript requires additional attention, including a better description of the novelty of the analysis and the interpretation of the study findings (e.g. correlational vs mechanistic).

Please take all of the reviewer comments into account in considering a revision of the manuscript.

Reviewer(s)' Comments to Author:

Referee: 1

Comments to the Author(s)

The study "Human footprint influences the global prevalence of a generalist parasite in mammalian wildlife" aims to explore variation in the infection of mammalian wildlife species with *Toxoplasma gondii* at global scale, compiling evidence from published field surveillance data and using a regression analysis, with host species traits, climate variables and a human footprint index as covariates.

I found this study a timely piece of work and a very well prepared manuscript. A possible shortfall, however, is in my opinion that the authors did not account for any possible spatial effect and sampling bias in their analysis. According to Figure 1, the underlying data are mostly clustered in N America and Europe, and I therefore think that such possible biases should be explored..

Perhaps the authors could also have put more efforts into distinguishing the different wildlife species in terms of their habitat use in addition to using an index of human food print in their analysis, as many wildlife species that are frequently recorded in anthropogenic habitats are known to share more pathogens with domestic animals and humans; this shortfall should at least be discussed if not included in an amended analysis.

There are multiple rows in the provided data table (file "MS_Data") with both zero for "pos" and "neg", which was unclear to me and needs clarification. Are the metadata for the data accessible?

Additional comments:

Line 37: I do not understand "ecological restoration" in context of your argument.

Line 55 and elsewhere: Given the importance of domestic cats in *Toxoplasma* transmission cycles, you should consider to extent your argument to the human-domestic animal-wildlife interface, in line with the growing number of studies exploring the importance of domestic and commensal animals for host sharing and pathogen spillover. Some recent study to incorporate on evidence of domestic species and anthropogenic habitat in pathogen host shifting and spillover;

Gibb, R., Redding, D.W., Chin, K.Q., Donnelly, C.A., Blackburn, T.M., Newbold, T. & Jones, K.E. (2020) Zoonotic host diversity increases in human-dominated ecosystems. *Nature*, 584, 398-402.

Wells, K., Morand, S., Wardeh, m. & Baylis, M. (2020) Distinct spread of DNA and RNA viruses among mammals amid prominent role of domestic species. *Global Ecology and Biogeography*, 29, 470-481.

Line 160: Its was unclear to me how you matched the terrestrial human footprint index to marine species such as Beluga or Dugongidae?

Line 201: Table 1 could be presented as supplementary material?

Lines 202-203: Did you test models with all different combination of variables? Your statement here to compare an intercept only model with one including ecological trait and climate variables leaves the question of how a model with human footprint HPF but not ecological trait and climate variable would compare to other models?

Lines 204-201: What do these posterior effects mean? Can they be expresses as odds ratio or % change in prevalence?

Lines 215-220: Perhaps worth to explore this phylogenetic effect further and mention any taxonomic groups with particularly high/low prevalence records?

Line 336: Please check if Dobson 2020 has been included in reference list.

Referee: 2

Comments to the Author(s)

General Comments

This is an interesting study in that it seeks to analyze factors related to *T. gondii* presence at a global scale. The power of the work is in the size of the database being analyzed. However, while you have a large database, it was somewhat unclear why this work is as important? In essence, the Introduction did not do a good job of indicating why the work you pursued was really needed beyond what we already know, especially if looking for factors that explain relationships. Aside from the lack of set up, my main issue with the work is that it presents itself as mechanistic in a way that is beyond the data. I know there is a divide between how spillover has been presented outside of ecology versus within the discipline, but the variables analyzed in this study really can only demonstrate if there are correlational relationships, not what are driving factors. As a result, there are not only a lot of caveats, but a number of definitive statements cannot be made. In addition, the lack of data on cats for your study sites mean that we cannot accurately test the relationship between HFI and *T. gondii*. All this is to say that I see value in the research, but that it is need of much better contextualization and stating what piece of knowledge it is adding to that we are lacking.

Specific Comments

L51. Suggest citing Chalkowski, K., C.A. Lepczyk, and S. Zohdy. 2018. Parasite ecology of invasive species: new directions. *Trends in Parasitology* 34:655–663.

L58. It is really unclear how this is related to *T. gondii* as haven't really described the linkages here.

L62. Provide citations here to support.

L62. Ok, but aside from mortality is behavior change. In both humans and wildlife there is quite a bit of data on how *T. gondii* changes animal behavior and this is something of great importance. I would suggest that you describe more about the parasite in terms of health implications as well. It isn't just mortality we worry about in humans.

L67. A big issue with wildlife is its impact on threatened and endangered species or species that are endemic. For instance, *T. gondii* is a large threat to many island animals.

L68. I don't see the connection to roads as written.

L73. Be specific and name house cats as the definitive host.

L86. Incomplete paragraph, need at least three sentences here.

L99. Anthropogenic?

L101. I wouldn't quite call a lot of this spillover. Also, consider how oocysts move in the environment, they can easily move in streams and across land in areas with rain and topography. For instance, see work by Van Wormer.

L104. This paragraph doesn't really connect to previous one. Also, I didn't see a set up prior to this paragraph as what the larger need is for your study. That is, what makes this work novel or needed?

L114. And temperature influence the number, survival, and prevalence of outdoor cats that have a large input of oocysts into the environment.

L128. Which form of Web of Science did you use as type matters. See Calver MC, Goldman B, Hutchings PA, Kingsford RT. 2017. Why discrepancies in searching the conservation biology literature matter. *Biological Conservation* 213:19-26.

L129. Why not also use 'T. gondii' variations?

L148. Need to describe how Elton Traits database categorizes trophic level numerically (what are values and what do they mean) so that reader can see link to your prediction.

L159. Why averages and not also minimums and maximums? Considering that both temp and precip factor into T. gondii distributions, wouldn't it have been important to evaluate these other measures as well, especially if trying to find which factor may be more important?

L160. Why use HFI over other types of measures, such as land cover, nighttime lights, pop density? I have some idea, but I also find that HFI can have some problems with it as you are lumping a lot of ideas into a single index. Need to justify.

L164. Buffer around what? Why 50 km? Also, need to describe the pixel size or minimum mapping unit of your GIS data as well as what datum and projections are being used.

L202. What are ecology variables?

L211. Move to Discussion.

L224. Please begin Discussion with answers to your predictions.

L226. I would disagree with this conclusion. Your work is entirely corollary and not mechanistic. A large missing component of your research is the number or density of cats in the environment or on the landscape. Essentially you are equating HFI with cat numbers, which may not always be the case. Thus, you may have found support for a prediction, but you have not really identified the mechanism here. I would certainly not call it anthropogenic spillover.

L243. This paragraph is very speculative. Some of these land cover types would also simply not have cats, so is it conversion versus fact that cats are less abundant in wetlands? In the end this paragraph comes across as more of a post hoc rationalization of the analyses as you did not look at land cover/use change or other variables discussed here. This is the tradeoff of using HFI.

L294. But how did those studies measure trophic levels? One problem with trait data is that different researchers use different values and not all of these studies may be comparable.

L321. Again, I would disagree with this point. You essentially have a large database that is more akin to biogeography research in which you can test patterns along a descriptive nature that support or refute hypotheses. But you cannot determine mechanisms here. This is a large misunderstanding of quite a bit of the spillover literature in terms of how landscape ecology, biogeography, and macroecology evaluate the relationships.

L328. Agree with all of this, but wasn't really the point of the ms.

Table 1. Did you begin with an explicit framework of a priori models to consider that were based on your predictions? If so, please list these models somewhere.

Figure 1. One item that is apparent from this figure is that there may be some notable bias in locations sampled. There are few areas that have a low HFI and also large numbers of biomes missing.

Author's Response to Decision Letter for (RSPB-2020-2551.R0)

See Appendix A.

RSPB-2021-0414.R0

Review form: Reviewer 2

Recommendation

Major revision is needed (please make suggestions in comments)

Scientific importance: Is the manuscript an original and important contribution to its field?

Good

General interest: Is the paper of sufficient general interest?

Good

Quality of the paper: Is the overall quality of the paper suitable?

Good

Is the length of the paper justified?

Yes

Should the paper be seen by a specialist statistical reviewer?

No

Do you have any concerns about statistical analyses in this paper? If so, please specify them explicitly in your report.

Yes

It is a condition of publication that authors make their supporting data, code and materials available - either as supplementary material or hosted in an external repository. Please rate, if applicable, the supporting data on the following criteria.

Is it accessible?

N/A

Is it clear?

N/A

Is it adequate?

N/A

Do you have any ethical concerns with this paper?

No

Comments to the Author

General Comments

Overall the manuscript has been greatly improved and I enjoyed reading it. Much of the revisions nicely addressed my previous comments. However, I have two larger issues that remain. First, while I fully understand the appeal of the HFI, I believe that using this as your main source of environmental data is limiting your ability to get at a more nuanced understanding of the relationships you are seeking to test. I would strongly encourage the use of different data to test your hypotheses. The second item is in relation to the distribution of your samples. You have a wonderful database, but it is important to consider where those were collected from and why. I am concerned that the spatial distribution of samples is biased towards certain types of environments or locations rather than where wildlife live or are abundant. This is not a criticism of the research goals, but it is important to consider that you may have disproportionate

sampling in some ecosystems or environments than others, simply based on why the original studies were conducted and these need to be considered.

Specific Comments

L1. Suggest that title be changed to remove 'influences' and replace with 'correlates' or 'relates to' as the research is correlational, not mechanistic.

L15. Please thank the anonymous reviewers.

L27. Change habitat to ecosystem as habitat is a species specific term. Change throughout ms as needed.

L35. Change 'influence' to 'related to' or 'correlate with.' Please revise all statements like this throughout ms.

L53. Change agriculture to livestock as these are different. Also, while some areas (e.g., Amazonia) are expanding for livestock, in many developed nations the issue is not extensification, but intensification, such as increased densities of pigs, chickens, and cows in smaller physical spaces. Also, the point here isn't extensification specifically, but density of the animals on that landscape. Please revise.

L57. Please remove quotes from around terms here and throughout ms. See Goldwasser 1998 Bulletin of Ecological Society of America.

L65. Delete 'extreme.'

L68. Just state felids as definitive hosts here. As written this sentence suggests many domestic and wild animals, which while true, could just as easily state that felids are the only group of species that are definitive hosts.

L78. Not always. Note that there are mammals that this does not occur. Thus, while toxo is found in nearly every animal it has been sought in, there are species that simply pass it through and it doesn't encyst.

L84. Change 'wildlife groups' to 'taxa.'

L98 & 102. Both sentences begin with same wording, please revise.

L104. Indicate you mean individuals here, not species.

L111. Replace 'evaluate' with 'test.'

L112. So is this one of your hypotheses? From here onward the ending of the Introduction is very confusing. I would suggest you combine all hypotheses and their respective predictions into a single concluding paragraph and make sure that indicates what you are hypothesizing or describe it numerically. For instance "We sought to test three hypotheses that relate to *T. gondii*. First...Second..."

L112-126. These should be described before you talk about hypotheses.

L126. What hypothesis does this prediction relate to? Predictions should follow on a hypothesis.

L129. Is this the second or third hypothesis?

L131-5. These would read more clearly if you state the prediction and then follow it by the rationale. As written they are awkwardly.

L142. Should also state if you followed the PRISMA procedure for conducting reviews. This should be provided as a supplement showing the numbers at each stage of process.

L152. Ok, but I would argue this is still a weak justification. Did you go through and calculate dispersal/movement based on an allometric equation by species? Given that you are dealing with such a wide variety of taxa your dispersal data are going to vary markedly. Thus, rather than an arm-waving description here you do need to calculate something like median and mean dispersal and perhaps group it by major taxa, and then demarcate your boundary. I know this is an extra step, but it is important that you have a fully justifiable choice for selecting a bound that is expected in landscape ecology and biogeography.

L155. How are anthropogenic pressures consistent over a 50 km boundary? What are you measuring as anthropogenic here, land cover, air pollution, etc.?

L168. Please indicate what Elton trait groups were merged into your foraging groups or give specific criteria on how percentages resulted in a species being defined as one of the four groups in your research. Essentially you need to tell the reader how if they went to Elton traits that their classification gets transitioned into yours so they could replicate.

L180. Comma after maximum.

L183. This gets back to my first review. HFI is a very mixed bag type of metric as it simply provides some score based on these other various inputs without expressly separating each. My problem, as I mention previously, is that the HFI does not include a lot of other measures of human influence, and is a very generic measure that lacks the kind of information that you are seeking to try to evaluate in your models. My concern is that this is an easy to access database that provides a single gross level metric that misses the nuances or background rationales for evaluating specific environmental attributes. While the database has value, I remain unconvinced here that this was the best one to use really go into depth on factors related to wildlife and disease. I would suggest redoing the analysis with land cover data and other separate databases that can give more support for specific relationships.

L211. I would suggest you specifically note which models or model sets are being used to test each hypothesis.

L234. This sentence doesn't make sense as written.

L241. Comma after species.

L245-9. Here it sounds like you broke apart the HFI, but that is not how you described it in Methods.

L265. But this should be completely expected as would not expect a lot of wildlife in strict agricultural lands and also I would expect that sampling effort for toxo is low in these areas. This brings up an important point of sample bias across ecosystems. If we use land cover a proxy for ecosystem, where are most of samples from? Forested locations, grasslands, urban areas? This is actually an issue that needs to be considered as the lack of relationships may simply be due to where the animals were sampled in the first place and hence strong inequities occur. Certainly for wild birds and mammals in the US, where we sample for diseases varies markedly.

L301. Yes, but how are your wildlife samples distributed across space? Are people sampling wildlife in any large numbers near farms? I think the problem here is several fold. Barn cats are localized and densities may be low, especially with increasing distances from farm. Also, what wild animals are on farms or near them? Many wildlife do occur in ag landscapes, but many do not. Third, seasonal mismatch. Even if wildlife are present are they in these locations at a time when they might interact with cats? Finally, sampling effort in these landscapes. How representative of a sample do you have in agricultural areas? All of these points lead me back to earlier comments about need to describe the spatial distribution of your samples across land cover or ecosystem types (if we don't have enough samples from some contexts, can we really test for relationships?) and also the lack of separate measures of the environment (rather than just HFI). As it currently stands this paragraph is

L310-15. I would suggest cutting this part as it is very speculative. Also, we know that species diversity declines with latitude and that there are many other ecological phenomena that have latitudinal gradients. Thus, without including such evaluations in your analysis, you really can't say more. I don't disagree that we find more cats in warmer areas, but you don't have data to say much aside from that you found a relationship.

L318. It isn't simply the average rainfall, but how much occurs per unit time, especially when topography is included. To move oocysts over land requires overland flow, which requires a larger volume of water in a short amount of time. All this is to say that frequency and amplitude of precipitation matter greatly.

L360. Also the strain/genotype of toxo matters. I think if your phylogenetic comparison doesn't include strain, that you are also missing a large component of possible explanatory power.

L363. Other possible/noted limits are (or maybe) unequal representation of samples across land cover/ecosystem type, and lack of strain/genotype information.

L388. These last two paragraphs, especially the last one, are not strongly related to the work.

They're good pieces of information and I don't disagree with them, but they make the Discussion longer and extend beyond what your work really was centered on.

L400. What are landscape ecosystem services?

Decision letter (RSPB-2021-0414.R0)

19-Mar-2021

Dear Dr Wilson:

I am writing to inform you that we have now obtained responses from referees on manuscript RSPB-2021-0414 entitled "Human density influences the prevalence of a generalist parasite in mammalian wildlife at the global scale." which you submitted to Proceedings B.

Unfortunately, your manuscript has been rejected following full peer review. Competition for space in Proceedings B is currently extremely severe, as many more manuscripts are submitted to us than we have space to print. We are therefore only able to publish those that are exceptional, convincing and present significant advances of broad interest, and must reject many good manuscripts.

Please find below the comments received from referees concerning your manuscript, not including confidential reports to the Editor. I hope you may find these useful should you wish to submit your manuscript elsewhere.

We are sorry that your manuscript has had an unfavourable outcome, but would like to thank you for offering your work to Proceedings B.

Sincerely,
Professor Hans Heesterbeek
mailto: proceedingsb@royalsociety.org

Associate Editor
Board Member
Comments to Author:

Following a substantial revision of the manuscript, concerns remain that were raised in the first round of review. These include limitations to using human footprint index as the main source of environmental data and insufficient discussion of potential biases in the spatial distribution of samples.

Reviewer(s)' Comments to Author:

Referee: 2

Comments to the Author(s).

General Comments

Overall the manuscript has been greatly improved and I enjoyed reading it. Much of the revisions nicely addressed my previous comments. However, I have two larger issues that remain. First, while I fully understand the appeal of the HFI, I believe that using this as your main source of environmental data is limiting your ability to get at a more nuanced understanding of the relationships you are seeking to test. I would strongly encourage the use of different data to test your hypotheses. The second item is in relation to the distribution of your samples. You have a wonderful database, but it is important to consider where those were collected from and why. I am concerned that the spatial distribution of samples is biased towards certain types of environments or locations rather than where wildlife live or are abundant. This is not a criticism of the research goals, but it is important to consider that you may have disproportionate sampling in some ecosystems or environments than others, simply based on why the original studies were conducted and these need to be considered.

Specific Comments

L1. Suggest that title be changed to remove 'influences' and replace with 'correlates' or 'relates to' as the research is correlational, not mechanistic.

- L15. Please thank the anonymous reviewers.
- L27. Change habitat to ecosystem as habitat is a species specific term. Change throughout ms as needed.
- L35. Change 'influence' to 'related to' or 'correlate with.' Please revise all statements like this throughout ms.
- L53. Change agriculture to livestock as these are different. Also, while some areas (e.g., Amazonia) are expanding for livestock, in many developed nations the issues is not extensification, but intensification, such as increased densities of pigs, chickens, and cows in smaller physical spaces. Also, the point here isn't extensification specifically, but density of the animals on that landscape. Please revise.
- L57. Please remove quotes from around terms here and throughout ms. See Goldwasser 1998 Bulletin of Ecological Society of America.
- L65. Delete 'extreme.'
- L68. Just state felids as definitive hosts here. As written this sentence suggests many domestic and wild animals, which while true, could just as easily state that felids are the only group of species that are definitive hosts.
- L78. Not always. Note that there are mammals that this does not occur. Thus, while toxo is found in nearly every animal it has been sought in, there are species that simply pass it through and it doesn't encyst.
- L84. Change 'wildlife groups' to 'taxa.'
- L98 & 102. Both sentences begin with same wording, please revise.
- L104. Indicate you mean individuals here, not species.
- L111. Replace 'evaluate' with 'test.'
- L112. So is this one of your hypotheses? From here onward the ending of the Introduction is very confusing. I would suggest you combine all hypotheses and their respective predictions into a single concluding paragraph and make sure that indicate what you are hypothesizing or describe it numerically. For instance "We sought to test three hypotheses that relate to *T. gondii*. First...Second..."
- L112-126. These should be described before you talk about hypotheses.
- L126. What hypothesis does this prediction relate to? Predictions should follow on a hypothesis.
- L129. Is this the second or third hypothesis?
- L131-5. These would read more clearly if you state the prediction and then follow it by the rationale. As written they are awkwardly.
- L142. Should also state if you followed the PRISMA procedure for conducting reviews. This should be provided as a supplement showing the numbers at each stage of process.
- L152. Ok, but I would argue this is still a weak justification. Did you go through and calculate dispersal/movement based on an allometric equation by species? Given that you are dealing with such a wide variety of taxa your dispersal data are going to vary markedly. Thus, rather than an arm-waiving description here you do need to calculate something like median and mean dispersal and perhaps group it by major taxa, and then demarcate your boundary. I know this is an extra step, but it is important that you have a fully justifiable choice for selecting a bound that is expected in landscape ecology and biogeography.
- L155. How are anthropogenic pressures consistent over a 50 km boundary? What are you measuring as anthropogenic here, land cover, air pollution, etc.?
- L168. Please indicate what Elton trait groups were merged into your foraging groups or give specific criteria on how percentages resulted in a species being defined as one of the four groups in your research. Essentially you need to tell the reader how if they went to Elton traits that their classification gets transitioned into yours so they could replicate.
- L180. Comma after maximum.
- L183. This gets back to my first review. HFI is a very mixed bag type of metric as it simply provides some score based on these other various inputs without expressly separating each. My problem, as I mention previously, is that the HFI does not include a lot of other measures of human influence, and is a very generic measure that lacks the kind of information that you are seeking to try to evaluate in your models. My concern is that this is an easy to access database that provides a single gross level metric that misses the nuances or background rationales for evaluating specific environmental attributes. While the database has value, I remain unconvinced

here that this was the best one to use really go into depth on factors related to wildlife and disease. I would suggest redoing the analysis with land cover data and other separate databases that can give more support for specific relationships.

L211. I would suggest you specifically note which models or model sets are being used to test each hypothesis.

L234. This sentence doesn't make sense as written.

L241. Comma after species.

L245-9. Here it sounds like you broke apart the HFI, but that is not how you described it in Methods.

L265. But this should be completely expected as would not expect a lot of wildlife in strict agricultural lands and also I would expect that sampling effort for toxo is low in these areas. This brings up an important point of sample bias across ecosystems. If we use land cover a proxy for ecosystem, where are most of samples from? Forested locations, grasslands, urban areas? This is actually an issue that needs to be considered as the lack of relationships may simply be due to where the animals were sampled in the first place and hence strong inequities occur. Certainly for wild birds and mammals in the US, where we sample for diseases varies markedly.

L301. Yes, but how are your wildlife samples distributed across space? Are people sampling wildlife in any large numbers near farms? I think the problem here is several fold. Barn cats are localized and densities may be low, especially with increasing distances from farm. Also, what wild animals are on farms or near them? Many wildlife do occur in ag landscapes, but many do not. Third, seasonal mismatch. Even if wildlife are present are they in these locations at a time when they might interact with cats? Finally, sampling effort in these landscapes. How representative of a sample do you have in agricultural areas? All of these points lead me back to earlier comments about need to describe the spatial distribution of your samples across land cover or ecosystem types (if we don't have enough samples from some contexts, can we really test for relationships?) and also the lack of separate measures of the environment (rather than just HFI). As it currently stands this paragraph is

L310-15. I would suggest cutting this part as it is very speculative. Also, we know that species diversity declines with latitude and that there are many other ecological phenomena that have latitudinal gradients. Thus, without including such evaluations in your analysis, you really can't say more. I don't disagree that we find more cats in warmer areas, but you don't have data to say much aside from that you found a relationship.

L318. It isn't simply the average rainfall, but how much occurs per unit time, especially when topography is included. To move oocysts over land requires overland flow, which requires a larger volume of water in a short amount of time. All this is to say that frequency and amplitude of precipitation matter greatly.

L360. Also the strain/genotype of toxo matters. I think if your phylogenetic comparison doesn't include strain, that you are also missing a large component of possible explanatory power.

L363. Other possible/noted limits are (or maybe) unequal representation of samples across land cover/ecosystem type, and lack of strain/genotype information.

L388. These last two paragraphs, especially the last one, are not strongly related to the work.

They're good pieces of information and I don't disagree with them, but they make the Discussion longer and extend beyond what your work really was centered on.

L400. What are landscape ecosystem services?

Author's Response to Decision Letter for (RSPB-2021-0414.R0)

See Appendix B.

RSPB-2021-1724.R0

Review form: Reviewer 2

Recommendation

Major revision is needed (please make suggestions in comments)

Scientific importance: Is the manuscript an original and important contribution to its field?

Excellent

General interest: Is the paper of sufficient general interest?

Good

Quality of the paper: Is the overall quality of the paper suitable?

Acceptable

Is the length of the paper justified?

Yes

Should the paper be seen by a specialist statistical reviewer?

Yes

Do you have any concerns about statistical analyses in this paper? If so, please specify them explicitly in your report.

Yes

It is a condition of publication that authors make their supporting data, code and materials available - either as supplementary material or hosted in an external repository. Please rate, if applicable, the supporting data on the following criteria.

Is it accessible?

Yes

Is it clear?

Yes

Is it adequate?

Yes

Do you have any ethical concerns with this paper?

No

Comments to the Author

General Comments

Overall the manuscript has been improved further from my previous evaluation of it. While improved, there remain a handful of items that need further attention. First, in terms of the statistical analyses and model building, now that you have clarified your three hypotheses and predictions, a bit more clarification is needed. Specifically, prior to model building you need to test your independent variables for collinearity and remove variables if needed (if you did this, then let reader know). In addition, while I understand the linear nature of the models, no reason was provided why only linear terms were used (i.e. no quadratic terms) or interactions. These may not at be needed, but it would be helpful to clarify here and if they should be considered, please do. Since you have three hypotheses, it would make more sense to have three separate model groupings that are being compared within a set. From Table 1 it appears that these are all

the models you ran, rather than model 1-5 are in relation to testing hypothesis 1, models 6-10 are in relation to testing to hypothesis 2, etc. Such a mirroring of analysis relative to hypothesis should be done. A final analytical point is that DIC interpretations are fine, but need to describe what level a model is no longer considered in top model set (similar to AIC >2) and also if cannot provide some measure of fit (e.g., akin to adjusted r²) it needs to be noted that you have top models, but that they might not describe much (this is an issue that readers need to be reminded of as AIC, DIC, etc., are valuable for informing models relative to one another, but they don't provide a measure of how much variance is described in an easy to interpret manner. This does not mean the work is problematic, rather it provides further contextualization of the research which is needed to make your findings crystal clear to readers).

Second, while I appreciate the authors view on bias, they are not necessarily accurate. This is not a point I wish to argue with the authors about, but having published widely with monitoring data and in macroecology there are numerous ways we treat data and account for it prior to statistical analyses that the authors have not done. Statistical analyses alone simply does not account for or removes bias. The points raised about lack of geographic coverage, effort, and time of year are all ways that the data may be biased that are not very well accounted for in the work. I consider the work in this manuscript as critically valuable and my points here are not to upset you the author as much as to make sure the work is bullet proof given the many controversies that any work involving cats entails. At a minimum all of these points need to be noted in the limitations paragraph of the Discussion.

Third, the argument about the area used as a landscape (50km²) skirted the issue quite a bit. For instance, you noted that for ~50% of the spp you could have found body size and estimated ranges. Yes, this would cut your spp sample size down, but the work would have greater ecological relevance by evaluating the landscape elements that lie within the dispersal distance of an average mammal (you could also have this analysis as a more in-depth approach and then used all spp in a more basic approach). I really was just not convinced that 50km is an ecologically justified distance (yes, dispersal and home range vary by season, location and spp, but we do know that on average a large mammal moves farther than a small mammal), especially with the diversity of mammals evaluated. Many macroecology studies do not simply use one scale, unless they have based it on something like similar spp or allometry. Minimally, it seems like an evaluation at several scales would have been done here or a subsetting of the data to look more closely at the spp for which you can find body mass data. The argument that other landscape ecology studies follow what you have done certainly occurs at times, but there is a strong literature pointing out that for landscape ecology to advance and be done well that it needs to have non-arbitrary bounding. The scale here is not based on animal movement, political boundary, etc.

Fourth, the Discussion should begin with answers to your hypotheses. Did you find support?

Fifth, the revised ms has a variety of spelling and grammatical items that should be revised (data are rather than data is, depredated instead of predate as predate means to occur before, short paragraphs that need to be combined with others as a paragraph should have at least three sentences, etc.).

Attending to these aspects will very much help improve the work in a manner that will advance our understanding of this parasite and its implications in a One Health context. I very much understand that my points require further work, but believe accounting for these considerations will yield a widely read and cited ms that has real world implications.

Decision letter (RSPB-2021-1724.R0)

31-Aug-2021

Dear Dr Wilson:

Your manuscript has now been peer reviewed and the review has been assessed by an Associate Editor. The reviewer's comments (not including confidential comments to the Editor) and the comments from the Associate Editor are included at the end of this email for your reference. As you will see, the reviewer and the Associate Editor have raised some issues to further improve your manuscript and we would like to invite you to revise your manuscript to address them.

We do not allow multiple rounds of revision so we urge you to make every effort to fully address all of the comments at this stage. If deemed necessary by the Associate Editor, your manuscript can be sent back to one or more of the original reviewers for assessment. If the original reviewers are not available we may invite new reviewers. Please note that we cannot guarantee eventual acceptance of your manuscript at this stage.

Research ethics:

Use of animals and field studies:

It is a condition of publication that you make available the data and research materials supporting the results in the article (<https://royalsociety.org/journals/authors/author-guidelines/#data>). Datasets should be deposited in an appropriate publicly available repository and details of the associated accession number, link or DOI to the datasets must be included in the Data Accessibility section of the article (<https://royalsociety.org/journals/ethics-policies/data-sharing-mining/>). Reference(s) to datasets should also be included in the reference list of the article with DOIs (where available).

Please submit a copy of your revised paper within three weeks. If we do not hear from you within this time your manuscript will be rejected. If you are unable to meet this deadline please let us know as soon as possible, as we may be able to grant a short extension.

Best wishes,
Professor Hans Heesterbeek
mailto:proceedingsb@royalsociety.org

Associate Editor
Comments to Author:

After reading this careful and extensive revision by Wilson et al., a reviewer has offered a few comments and suggestions. In preparing a response to reviewer comments, please indicate further edits that have been made to the manuscript to address these remaining suggestions or please offer explanation for when there is a difference of opinion with the reviewer. Some additional methodological details also may benefit the manuscript, including details on GIS/imagery data (projection and datum used, size of pixels or minimum mapping unit, when data are from) and details on editing/prepping the data, evaluating it for use in a GLM, and details on model building.

Reviewer(s)' Comments to Author:

Referee: 2

Comments to the Author(s).

General Comments

Overall the manuscript has been improved further from my previous evaluation of it. While improved, there remain a handful of items that need further attention. First, in terms of the statistical analyses and model building, now that you have clarified your three hypotheses and predictions, a bit more clarification is needed. Specifically, prior to model building you need to test your independent variables for collinearity and remove variables if needed (if you did this, then let reader know). In addition, while I understand the linear nature of the models, no reason was provided why only linear terms were used (i.e. no quadratic terms) or interactions. These may not at be needed, but it would be helpful to clarify here and if they should be considered,

please do. Since you have three hypotheses, it would make more sense to have three separate model groupings that are being compared within a set. From Table 1 it appears that these are all the models you ran, rather than model 1-5 are in relation to testing hypothesis 1, models 6-10 are in relation to testing to hypothesis 2, etc. Such a mirroring of analysis relative to hypothesis should be done. A final analytical point is that DIC interpretations are fine, but need to describe what level a model is no longer considered in top model set (similar to $AIC > 2$) and also if cannot provide some measure of fit (e.g., akin to adjusted r^2) it needs to be noted that you have top models, but that they might not describe much (this is an issue that readers need to be reminded of as AIC, DIC, etc., are valuable for informing models relative to one another, but they don't provide a measure of how much variance is described in an easy to interpret manner. This does not mean the work is problematic, rather it provides further contextualization of the research which is needed to make your findings crystal clear to readers).

Second, while I appreciate the authors view on bias, they are not necessarily accurate. This is not a point I wish to argue with the authors about, but having published widely with monitoring data and in macroecology there are numerous ways we treat data and account for it prior to statistical analyses that the authors have not done. Statistical analyses alone simply does not account for or removes bias. The points raised about lack of geographic coverage, effort, and time of year are all ways that the data may be biased that are not very well accounted for in the work. I consider the work in this manuscript as critically valuable and my points here are not to upset you the author as much as to make sure the work is bullet proof given the many controversies that any work involving cats entails. At a minimum all of these points need to be noted in the limitations paragraph of the Discussion.

Third, the argument about the area used as a landscape (50km²) skirted the issue quite a bit. For instance, you noted that for ~50% of the spp you could have found body size and estimated ranges. Yes, this would cut your spp sample size down, but the work would have greater ecological relevance by evaluating the landscape elements that lie within the dispersal distance of an average mammal (you could also have this analysis as a more in-depth approach and then used all spp in a more basic approach). I really was just not convinced that 50km is an ecologically justified distance (yes, dispersal and home range vary by season, location and spp, but we do know that on average a large mammal moves farther than a small mammal), especially with the diversity of mammals evaluated. Many macroecology studies do not simply use one scale, unless they have based it on something like similar spp or allometry. Minimally, it seems like an evaluation at several scales would have been done here or a subsetting of the data to look more closely at the spp for which you can find body mass data. The argument that other landscape ecology studies follow what you have done certainly occurs at times, but there is a strong literature pointing out that for landscape ecology to advance and be done well that it needs to have non-arbitrary bounding. The scale here is not based on animal movement, political boundary, etc.

Fourth, the Discussion should begin with answers to your hypotheses. Did you find support?

Fifth, the revised ms has a variety of spelling and grammatical items that should be revised (data are rather than data is, depredated instead of predate as predate means to occur before, short paragraphs that need to be combined with others as a paragraph should have at least three sentences, etc.).

Attending to these aspects will very much help improve the work in a manner that will advance our understanding of this parasite and its implications in a One Health context. I very much understand that my points require further work, but believe accounting for these considerations will yield a widely read and cited ms that has real world implications.

Author's Response to Decision Letter for (RSPB-2021-1724.R0)

See Appendix C.

Decision letter (RSPB-2021-1724.R1)

23-Sep-2021

Dear Dr Wilson

I am pleased to inform you that your manuscript entitled "Human density is associated with the increased prevalence of a generalist zoonotic parasite in mammalian wildlife" has been accepted for publication in Proceedings B.

Data Accessibility section

Open Access

You are invited to opt for Open Access, making your freely available to all as soon as it is ready for publication under a CCBY licence. Our article processing charge for Open Access is £1700. Corresponding authors from member institutions (<http://royalsocietypublishing.org/site/librarians/allmembers.xhtml>) receive a 25% discount to these charges. For more information please visit <http://royalsocietypublishing.org/open-access>.

Paper charges

Sincerely,

Professor Hans Heesterbeek

Associate Editor:

Board Member

Comments to Author:

The authors deserve credit for their efforts to address the many comments and questions that arose during multiple rounds of review. Their manuscript will make a strong contribution to the macroecology literature investigating environmental factors that determine the prevalence of parasites associated with wildlife.

Appendix A

THE UNIVERSITY OF BRITISH COLUMBIA

Department of Forest and Conservation Sciences

3041-2424 Main Mall
Vancouver, BC V6T 1Z4
Tel: (604) 822-9695
Fax: (604) 822-9102

February 17, 2021

Dear Professors Barrett and Heesterbeek,

Re: Resubmission of RSPB-2020-2551

We thank you very much for offering us the opportunity to submit a revised version of our manuscript for the Proceedings of the Royal Society B. We have put a considerable effort into revising the manuscript following the helpful comments from yourself and the two reviewers. Below, we have numbered all of the reviewer comments and provided our response underneath each, along with italicized text where possible to include text excerpts.

Here, we would like to highlight the more substantial changes we made to the manuscript and then these are described in more detail where they correspond to the reviewers' comments. Based on the request from reviewer one, we have now provided prevalence estimates along with confidence intervals for all taxonomic families. We have also calculated probabilities of *T. gondii* infection as a function of human density both as text and as a figure.

Based on the comments from reviewer two, we reanalyzed the data with additional non-composite anthropogenic variables, crop pressure and human density. We chose these variables because of the available geographic databases and that crop pressure and human density represent agricultural and urban landscapes respectively, and therefore, allowed us

to consider whether the type of anthropogenic landscape influenced *T. gondii* epidemiology. Refining our analysis allowed us to show that human density pressure but not agricultural expansion is an influential anthropogenic driver of *T. gondii* prevalence.

Reviewer two also asked us to highlight the novelty of this work further. *Toxoplasma gondii* is unique in its broad host range and also in the extensive amount of available surveillance data, which allowed us to employ the macroecological approach taken in this paper. Several studies have correlated *T. gondii* prevalence with geographical variables over small spatial scales, often qualitatively. Our study is unique as we are analyzing georeferenced data across both climatic and anthropogenic gradients from a wide range of species, which is a much more robust test of a variable's influence. Our approach to quantitatively control for phylogenetic distance is also infrequently done but provides important cues for when phylogenetically conserved ecological or physiological traits may be influential as compared to geographical variables. In this restructured manuscript, we more clearly articulate the novelty of this work throughout the text, especially in the Introduction.

Finally, although the positive influence of human density pressure lends further support to the important role of domestic cats as a driver of *T. gondii* prevalence, we have explicitly discussed the correlational nature of our results and the limitations of our interpretation.

Below we elaborate on these points as well as the many others made by yourself and the reviewers. We believe our manuscript is much improved following these revisions, and we look forward to your decision.

Sincerely,

Amy Wilson on behalf of co-authors

Reviewer(s)' Comments to Author:

Referee: 1

Comments to the Author(s)

R1, Comment 1: The study “Human footprint influences the global prevalence of a generalist parasite in mammalian wildlife” aims to explore variation in the infection of mammalian wildlife species with *Toxoplasma gondii* at global scale, compiling evidence from published field surveillance data and using a regression analysis, with host species traits, climate variables and a human footprint index as covariates.

I found this study a timely piece of work and a very well prepared manuscript. A possible shortfall, however, is in my opinion that the authors did not account for any possible spatial effect and sampling bias in their analysis. According to Figure 1, the underlying data are mostly clustered in N America and Europe, and I therefore think that such possible biases should be explored.

Response: We acknowledge that research effort is concentrated in particular geographic regions, but the parameters of interest are still well sampled within those regions (e.g. a gradient from zero footprint to very high footprint). Therefore, we are confident in the relationships we found but agree with the reviewer that a caveat is needed to acknowledge that some global regions are undersampled, and further study is needed to determine if the relationships we found also hold in those regions (e.g. central Asia). We believe it is likely that they will because the parameters we surveyed vary across similar gradients in those regions (e.g. footprint varies similarly across parts of central Asia as it does in Europe) however, this is still important to acknowledge. We have highlighted the sampling disparity on lines 362-369:

*We acknowledge two important caveats in our study. First, although a large amount of *T. gondii* prevalence data was available, there are key areas such as central Eurasia and Africa that had minimal representation, which is unfortunate given that countries on these continents have relatively high human *T. gondii**

prevalence [12]. Therefore, the associations found in the present analysis may not be as influential for populations in those data deficient regions. Increasing surveillance of wildlife for underrepresented taxa and regions would enable a more refined analysis for ecological and geographical variables that may influence *T. gondii* prevalence.

R1, Comment 2: Perhaps the authors could also have put more efforts into distinguishing the different wildlife species in terms of their habitat use in addition to using an index of human food print in their analysis, as many wildlife species that are frequently recorded in anthropogenic habitats are known to share more pathogens with domestic animals and humans; this shortfall should at least be discussed if not included in an amended analysis.

Response: Due to geographic variation in habitat use relative to availability, and the absence of reliable data for all the species, we elected to use broad categories for the analysis since we had specific predictions on terrestrial versus aquatic habitat associations. We are unaware of any published material that has indices of associations with human environments for all of the species in this analysis. However, to check on this possibility in preparing the revision, we did an additional analysis. The Pantheria database, a large compilation of ecological information for all mammal species, provides a measure of the average human density over the range of species. This human density measure was available for 209 of the 238 species (N= 39,027). We used this as a species-level trait in our analyses to test if this measure of synanthropism would be influential for *T. gondii* prevalence. We did not find any support for the inclusion of this variable and the results were consistent in the significance of habitat (-1.13 (95% CI: -1.9 - -0.25)), average temperature (0.34 (95% CI: 0.08 -0.58)), and population density (0.28 (95% CI: 0.04 - 0.51)). This is shown in the following table (not in manuscript).

Review Comment Table 1. Deviance Information Criterion (DIC) model selection results testing hypotheses for ecological, environmental and anthropogenic risk factors influencing *Toxoplasma gondii* prevalence for a subset of the data where species-level synanthropism metric was available. Δ DIC = change in DIC relative to top model

Model variables	DIC	ΔDIC
Habitat + diet + temp + ppt + human density	27470.36	0.00
Habitat + diet + synathr + temp + ppt + human density	27471.21	0.85
Habitat + diet + synathr+ temp + ppt	27473.93	3.57
Habitat + diet + synathr + temp + ppt + crop density	27474.27	3.91

We elected not to include this analysis since we found no evidence for an effect and because our model included taxonomic representation that would help control for any inherent phylogenetic effects (e.g. if some taxa were inherently more susceptible). However, we have still discussed this concept more generally in the text on lines 342-346, where we refer to previous studies that have found that synanthropism increases disease exposure risk in general and give one example of two closely related species where the one that has higher associations with human environments has higher prevalence:

Lines 342-346 now read:

The results from this study and other large-scale analyses (Murray et al. 2019) highlight how anthropogenic pressures in the habitat can modify disease risk irrespective of ecological traits. For example, within this dataset, black bears (Ursus americanus) had double the prevalence (58 %) of the less synanthropic congeners, but similarly omnivorous, grizzly bears (U. arctos, 24 %).

We are certainly happy to consider options for this topic further if desired by the reviewer or editor.

R1, Comment 3: There are multiple rows in the provided data table (file “MS_Data”) with both zero for “pos” and “neg”, which was unclear to me and needs clarification. Are the metadata for the data accessible?

Response: This was a file conversion error and has been corrected. We have also included a sheet in the excel file along with the dataset that explains the variables.

Additional comments:

R1, Comment 4: Line 37: I do not understand “ecological restoration” in context of your argument.

Response: we have adjusted this to better clarify and line 39-41 now reads:

“...Reducing the environmental input of *T. gondii* from domestic cat sources would synergistically reduce wildlife and human infection risks from one of the world's most common parasitic infections..”

R1, Comment 5: Line 55 and elsewhere: Given the importance of domestic cats in Toxoplasma transmission cycles, you should consider to extent your argument to the human-domestic animal-wildlife interface, in line with the growing number of studies exploring the importance of domestic and commensal animals for host sharing and pathogen spillover. Some recent study to incorporate on evidence of domestic species and anthropogenic habitat in pathogen host shifting and spillover;

Gibb, R., Redding, D.W., Chin, K.Q., Donnelly, C.A., Blackburn, T.M., Newbold, T. & Jones, K.E. (2020) Zoonotic host diversity increases in human-dominated ecosystems. Nature, 584, 398-402.

Wells, K., Morand, S., Wardeh, m. & Baylis, M. (2020) Distinct spread of DNA and RNA viruses among mammals amid prominent role of domestic species. Global Ecology and Biogeography, 29, 470-481.

Response: These were very helpful citations and aided us in the introduction (lines 50, 55-58), and we reemphasize this important point in the concluding paragraph (391-393).

Line 50: “...conditions are: altering host community structure [1,3]..”

Lines 55-58 now read:

Although wildlife populations have increased pathogen richness, it is domesticated animals that have a more central role in ‘sharing’ generalist pathogens with both humans and wildlife, ranging from viruses [7] to helminths [8] and ectoparasites [9].

Lines 391-393 now read:

Therefore, the intermingling of domestic cats with wildlife acts as problematic conduit for disease transmission across the wildlife-domestic animal-human interfaces and should be limited.

R1, Comment 6: Line 160: Its was unclear to me how you matched the terrestrial human footprint index to marine species such as Beluga or Dugongidae?

Response: We had included the Dugong in the original version as we felt that the sampling location would approximate the exposure location but further literature review of their dispersal ecology revealed that Dugong disperse large distances, so we removed it from the data and reanalysed the data in the revision. We rechecked the data and did not have Beluga (*Delphinapterus leucas*) in the dataset but do discuss pertinent studies for Beluga (line 328).

R1, Comment 7: Line 201: Table 1 could be presented as supplementary material?

Response: We had initially placed Table 1 in the general text so that readers could immediately read the comparative data, and we have left it in the text here. However, we could place the table in the supplementary material section if preferred.

R1, Comment 8: Lines 202-203: Did you test models with all different combination of variables? Your statement here to compare an intercept only model with one including ecological trait and climate variables leaves the question of how a model with human footprint HPF but not ecological trait and climate variable would compare to other models?

Response: For our analyses, we used a hypothesis testing approach where we evaluated the support for *T. gondii* prevalence being influenced by climatic, anthropogenic and ecological effects. In the first revision, we did this in a stepwise manner with sequentially adding variables and interpreting the resulting model support. In our reanalysis of the data for the revision, we modified our approach to address this comment. We considered the

climatic and ecological models separately and only retained significant variables (where significance was interpreted as variables whose coefficient does not include zero) in the subsequent models where anthropogenic variables were tested. Since temperature and habitat are influential variables explaining *T. gondii* prevalence, their inclusion improves model support and did not change the significance of the human density pressure.

Although not shown in the manuscript, model containing only human density pressure (with method and random effects) had a DIC of 30571, which was a Δ DIC of 5 higher than the more supported model with habitat, temperature and human density pressure. We interpreted the influence of a variable both by the improvement of model fit (DIC) with the inclusion of that variable and also the posterior mean (β) of the variable in the resulting model where the confidence intervals excluded 0.

R1, Comment 9: Lines 204-201: What do these posterior effects mean? Can they be expressed as odds ratio or % change in prevalence?

Response: The posterior mean effects are the beta coefficients, and in this case, represent the coefficient for each variable on a logit scale. We recognize that logit scale coefficients are not as intuitive and therefore to address this comment further, we used the top model to calculate the predicted probability of *T. gondii* infections for terrestrial and aquatic taxa across the range of human density pressures observed in the data, while maintaining temperature at the mean value. We discuss this increased risk in terms of percentage in lines 250-253 (below).

Based on this top model, the mean prevalence at the minimum and maximum human density pressures observed in this study would be three times higher both for terrestrial (0.05 to 0.16) and aquatic taxa (0.11 to 0.33) at mean temperatures.

R1, Comment 10: Lines 215-220: Perhaps worth to explore this phylogenetic effect further

Response: One of the more novel aspects of our analysis is that we did quantitatively control for phylogenetic distance, so it was an oversight not to more fully describe this

result. We expand upon the phylogenetic signal in lines 348-360 of the Discussion, and the text now reads:

*We quantitatively controlled for the phylogenetic distance between taxa in our analyses, thus enabling us to calculate a phylogenetic signal. Although few studies have quantified a phylogenetic signal for wildlife diseases which limits comparison, the signal for *T. gondii* was higher than the signal reported for a smaller scale study focused on avian malaria among multiple orders of Andean birds (λ : 0.13 – 0.35; [63]). The phylogenetic signal for a generalist pathogen like *T. gondii* likely reflects evolutionarily-conserved host ecological or physiological traits that influenced prevalence, but were not included in our global analysis. For example, the evolutionarily distant families, Didelphidae and Procyonidae are both synanthropic, terrestrial omnivores, but the former has almost half the prevalence (supplementary table S3), suggesting the presence of other unaccounted for drivers. Clinical severity of *T. gondii* infections does vary among taxonomic groups, with related taxa showing similar patterns of susceptibility, which would also be a source for the phylogenetic signal in this dataset [63].*

R1, Comment 11: Mention any taxonomic groups with particularly high/low prevalence records?

Response: This was an important recommendation for the readers so in this revision, we discuss taxonomic comparisons in lines 343-345 and 354-359. We also took this recommendation slightly further and calculated family level prevalence estimates corrected for sampling effort in supplementary table S2.

Lines 344-346:

*For example, within this dataset, black bears (*Ursus americanus*) had double the prevalence (58 %) of the less synanthropic, but similarly omnivorous, grizzly bears (*U. arctos*, 24 %).*

Lines 355-360:

For example, the evolutionarily distant families, Didelphidae and Procyonidae are both synanthropic, terrestrial omnivores, but the former has almost half the prevalence (supplementary table S3), suggesting the presence of other unaccounted for drivers. Clinical severity of T. gondii infections does vary among taxonomic groups, with related taxa showing similar patterns of susceptibility, which would also be a source for the phylogenetic signal in this dataset [63].

R1, Comment 12: Line 336: Please check if Dobson 2020 has been included in reference list.

Response: We had omitted this reference from the citation list and have now corrected this error, and the Dobson et al 2020 citation is now included in line 640.

Referee: 2

Comments to the Author(s)

General Comments

R2, Comment 1: This is an interesting study in that it seeks to analyze factors related to *T. gondii* presence at a global scale. The power of the work is in the size of the database being analyzed. However, while you have a large database, it was somewhat unclear why this work is as important? In essence, the Introduction did not do a good job of indicating why the work you pursued was really needed beyond what we already know, especially if looking for factors that explain relationships.

Response: Thank you for emphasizing this point, we realize that we needed to further detail the novelty and value of this study in the Introduction and we have rewritten those sections to do so. In particular, the size of the database that we compiled enabled us to test hypotheses on the influence of anthropogenic, climatic, ecological and taxonomic factors on *T. gondii* prevalence in a much more substantive way than finer scale studies that will have had a much narrower range in these variables. This allowed us to draw broad conclusions on how these factors influence prevalence across a large range of mammalian taxa and at a near global scale. We put a substantial amount of effort into harvesting data and obtaining locations to create a dataset that would permit a quantitative test of these drivers across mammalian taxa and under different combinations of climate and anthropogenic gradients, which thus far has not been done. However, we agree that someone unfamiliar with these studies would not realize this, and so we have expanded on this in the Introduction. For example, lines 101-109 now read:

*Due to these public and wildlife health impacts, considerable research effort has been directed towards measuring *T. gondii* prevalence in wildlife, with data being available for tens of thousands of wild animals worldwide. However, characterizing the probable drivers of these prevalence patterns has received far less attention. Since there is surveillance data representative of a range of gradients in different climates for diverse taxonomic groups, it is possible to*

collectively use these studies to disentangle climatic, anthropogenic, ecological and phylogenetic correlates of disease prevalence. Moreover, the broad host range, coupled with the extensive survey effort of T. gondii, enables a global perspective that can rarely be accomplished for pathogens.

R2, Comment 2: Aside from the lack of set up, my main issue with the work is that it presents itself as mechanistic in a way that is beyond the data. I know there is a divide between how spillover has been presented outside of ecology versus within the discipline, but the variables analyzed in this study really can only demonstrate if there are correlational relationships, not what are driving factors.

As a result, there are not only a lot of caveats, but a number of definitive statements cannot be made. In addition, the lack of data on cats for your study sites mean that we cannot accurately test the relationship between HFI and *T. gondii*. All this is to say that I see value in the research, but that it is need of much better contextualization and stating what piece of knowledge it is adding to that we are lacking.

Response: We have addressed Reviewer two's concern by both reframing our interpretation of cat densities as a driver of *T. gondii* prevalence more cautiously (lines 268-270) and explicitly discussed the limitations of using correlative data for inferring causation (lines 368-371).

Lines 268-270:

Although such a direct association cannot be made at the global scale due to a lack of global cat density data, domestic cat abundance is intrinsically linked to human presence.

Lines 369-372:

Second, while our model can help predict global trends in T. gondii associated with climate and human densities, we acknowledge that identification of cause and

effect mechanisms, such as domestic cat abundance, can only be deduced in a general manner at a global spatial scale.

The revision did include a complete reanalysis of the data and in addition to the human footprint included, an assessment of the human density and crop pressures, which are likely to reflect cat densities more closely (especially human density, given our close association with domestic cats). We believe that our approach did allow us to test the relationship between anthropogenic variables and *T. gondii* because we used a robust quantitative analysis, but we agree that we do not have data on cat densities and that there are other factors influencing *T. gondii* prevalence as we showed for temperature and terrestrial versus aquatic habitat associations.

Due to the difficulty of estimating cat densities over larger spatial scales, with limited to no data from many parts of the world, we feel that human density pressure is perhaps the most tractable proxy for cat population density, which we hope the reviewers agree with. That being said, we would be remiss not to fully discuss that domestic cats would be a key contributor to the positive correlation between *Toxoplasma gondii* and human density because felids are the ultimate source of *Toxoplasma gondii* and domestic cat density are orders of magnitude higher than densities of native cats. For example, British Columbia has one of the world's highest cougar populations, estimated to be 4,000-6,000 individuals (1). Humane Canada (2) suggests that the average Canadian household owns 1.6 cats, which potentially puts the owned cat population in British Columbia into the millions, without even considering the feral populations. On Vancouver Island, where the only two felids are cougars and domestic cats, cougars could be outnumbered by 1000 fold. Therefore, the link to cats is of crucial importance to understand the epidemiology of a pervasive pathogen, and serve as another example where unmanaged domestic animals can be important sources of disease for both humans and wildlife. Prominent studies have emphasized the risk of zoonotic disease from wildlife using the diversity of pathogens and fail to account for the actual human health burden, for which domestic animals are greater contributors.

(1) Cougar Management Guidelines Working Group (2005) Cougar management guidelines. WildFutures, Bainbridge Island, Washington, USA.

(2) Humane Canada – Cats in Canada. Available: https://humanecanada.ca/wp-content/uploads/2020/03/Cats_In_Canada_ENGLISH.pdf

Specific Comments

R2, Comment 3: L51. Suggest citing Chalkowski, K., C.A. Lepczyk, and S. Zohdy. 2018. Parasite ecology of invasive species: new directions. *Trends in Parasitology* 34:655–663.

Response: This was a very interesting review article and we have included it in our introduction in lines 49-52:

Examples of anthropogenic mechanisms are: altering host community structure (Keesing et al. 2010; Gibb et al. 2020), reducing host health and immunity (Murray et al. 2019), disrupting ecosystem services capable of reducing pathogen transmission (Gottdenker et al. 2014; Shapiro et al. 2010) and invasive species and pathogen introduction (Chalkowski, Lepczyk, and Zohdy 2018).

R2, Comment 4: L58. It is really unclear how this is related to *T. gondii* as haven't really described the linkages here.

Response: Agreed, we have removed the livestock reference and now refer to domestic animals, and the following sentence also provides additional relevant background. Line 52-58 now reads:

Agricultural extensification, urbanization and globalization further amplify transmission risk by creating large, high-density domestic animal populations that can act as amplifying hosts at human-domestic animal-wildlife interfaces [1]. Although wildlife populations have increased pathogen richness, it is domesticated animals that have a more central role in 'sharing' generalist pathogens with both humans and wildlife, ranging from viruses [7] to helminths [8] and ectoparasites [9].

R2, Comment 5: L62. Provide citations here to support.

Response: This line was removed in the revision.

R2, Comment 6: L62. Ok, but aside from mortality is behavior change. In both humans and wildlife there is quite a bit of data on how *T. gondii* changes animal behavior and this is something of great importance. I would suggest that you describe more about the parasite in terms of health implications as well. It isn't just mortality we worry about in humans.

Response: We had discussed sources of morbidity but understand how this section was too brief. In response to this recommendation, we have expanded this section considerably and discuss the chronic health impacts of *T. gondii* including the behavioural modifications.

Lines 85 - 93 now read:

However, even for healthy or insensitive hosts, latent T. gondii tissue cysts within neurological and muscular tissue can cause subclinical effects for the host's lifespan [17]. In humans, latent toxoplasmosis has identified associations with severe mental illnesses (e.g. schizophrenia), epilepsy, autism, cognition, vision deficits, cancers, traffic accidents and increased severity of other current diseases [12,19]. Infected hosts can also have subtle but impactful behavioural changes such as reduced vigilance or even an attraction to felids, which for felid prey species would serve to complete the sexual lifecycle of the parasite. These behavioural changes have been documented in species ranging from rats [20] to chimpanzees [21].

R2, Comment 7: L67. A big issue with wildlife is its impact on threatened and endangered species or species that are endemic. For instance, *T. gondii* is a large threat to many island animals.

Response: We have now discussed this in the introduction in lines 82-85.

Toxoplasma gondii infections can be fatal for immunocompromised humans or animals [17]. Multiple wildlife groups are susceptible to *T. gondii* infection, such as marine mammals, Australian marsupials and New World monkeys, with reported mortalities in numerous endangered and endemic species [18].

R2, Comment 8: L68. I don't see the connection to roads as written.

Response: We meant to convey the increase in roadkill, so have expanded the text. Lines 93-97 now read:

*Although poorly understood, latent *T. gondii* infections in wildlife have been associated with reduced population fitness through altered fetal development (Formenti et al. 2015), increased mortality due to automobile collisions (Hollings et al. 2013), cold weather (Jokelainen et al. 2011) and concurrent parasitic infections (Gibson et al. 2011).*

R2, Comment 9: L73. Be specific and name house cats as the definitive host.

Response: Change made. Line 70-71 now reads:

Domestic cats and wild felids serve as the definitive hosts in the domestic and sylvatic cycles respectively.

R2, Comment 10: L86. Incomplete paragraph, need at least three sentences here.

Response: During the revision, this paragraph was removed and the information was integrated elsewhere.

R2, Comment 11:L99. Anthropogenic?

Response: This sentence was removed during revision.

R2, Comment 12: L101. I wouldn't quite call a lot of this spillover. Also, consider how oocysts move in the environment, they can easily move in streams and across land in areas with rain and topography. For instance, see work by Van Wormer.

Response: We have adjusted our use of the term throughout the manuscript. We were liberally considering the movement of the domestic strain of *T. gondii* into wildlife as being as if it was a pathogen into a novel host, which we have now changed. We refer to the movement of oocysts in lines 73-75.

These oocysts can survive for years in soil or water (Frenkel, Ruiz, and Chinchilla 1975; Lindsay and Dubey 2009), dispersing passively through the soil matrix and contaminating aquatic ecosystems via surface runoff (Shapiro et al. 2019).

And remind the reader in lines 123-125:

"...while heavy precipitation facilitates oocyst dispersal through terrestrial habitats and the transfer to aquatic habitats (Simon et al. 2013; VanWormer et al. 2013)."

R2, Comment 13: L104. This paragraph doesn't really connect to previous one. Also, I didn't see a set up prior to this paragraph as what the larger need is for your study. That is, what makes this work novel or needed?

Response: We have restructured the introduction considerably to improve our conveyance of the novelty of this work. This is discussed in particular in lines 101-109, which now read:

*Due to these public and wildlife health impacts, considerable research effort has been directed towards measuring *T. gondii* prevalence in wildlife, with data being available for tens of thousands of wild animals worldwide. However, characterizing the probable drivers of these prevalence patterns has received far less attention. Since there is surveillance data representative of a range of gradients in different climates for diverse taxonomic groups, it is possible to collectively use these studies to disentangle climatic, anthropogenic, ecological and phylogenetic correlates of disease prevalence. Moreover, the broad host*

range, coupled with the extensive survey effort of T. gondii, enables a global perspective that can rarely be accomplished for pathogens.

R2, Comment 14: L114. And temperature influence the number, survival, and prevalence of outdoor cats that have a large input of oocysts into the environment.

Response: We had discussed how temperature may influence outdoor cat activity in lines 121-126, which now read:

Local climatic factors are also necessary to consider since temperature and precipitation affect oocyst survival and dispersal and domestic cat abundance and activity (Gilot-Fromont et al. 2012), so we also evaluated the association between average temperature and precipitation on T. gondii prevalence.

This point is reiterated again in the discussion in lines 312-315, which read:

Therefore, we suspect that reduced T. gondii prevalence in colder habitats results from the smaller domestic cat populations because harsh winters would limit feral cat survival, and free-roaming of owned cats would be more seasonal.

R2, Comment 15: L128. Which form of Web of Science did you use as type matters. See Calver MC, Goldman B, Hutchings PA, Kingsford RT. 2017. Why discrepancies in searching the conservation biology literature matter. *Biological Conservation* 213:19-26.

Response: We thank the reviewer for bringing to our attention this interesting comparative study. For this study, we used Web of Science Core Collection and not the more restrictive Web of Science. We now state that in lines 141-142. We also used the citation lists of these studies to locate papers that we had not encountered (stated in lines 143-144). Based on Calver et al. (2017), WoSCC had some limitations compared to Google Scholar in terms of the number of studies found but using WoSCC and citation tracing, we were able to locate several hundred studies. In addition, we did limit the studies to peer-reviewed literature.

R2, Comment 16: L129. Why not also use ‘T. gondii’ variations?

Response: We did not think to use that as a search term as we assumed that the keyword used would not be truncated. We feel that the use of the search term “*Toxoplasma gondii*” would generally encompass the vast majority of studies that were detected using “T. gondii”. Upon testing this with WoSCC, the species name ‘gondii’ seems to be used in the search term so we would expect to capture any papers with the term *Toxoplasma gondii* even if truncated, but not vice versa.

R2, Comment 17: L148. Need to describe how Elton Traits database categorizes trophic level numerically (what are values and what do they mean) so that reader can see link to your prediction.

Response: We have provided additional details on lines 164-172 which now read:

The trophic level of a species was based on the proportion of a species’ diet consisting of fruit, nectar, seeds, invertebrates, vertebrates and vertebrate prey as provided in EltonTraits 1.0 [38]. We reclassified these percentages into four categories: herbivores, invertivores, omnivores and carnivores. Herbivores and invertivores were species where the highest percentage of the diet was vegetation and invertebrate prey, respectively. We defined omnivores as species where the primary diet could be vegetation or invertebrates but included <50% vertebrates. Carnivores were species that primarily consumed >50% live vertebrate prey or carrion.

R2, Comment 18: L159. Why averages and not also minimums and maximums?

Considering that both temp and precip factor into T. gondii distributions, wouldn’t it have been important to evaluate these other measures as well, especially if trying to find which factor may be more important?

Response: In exploratory analyses, we did test minimum and maximum temperatures and found that average temperature was more influential. We now state this in lines 179-182, which now read:

*The average, maximum and minimum temperatures were highly correlated, and exploratory analyses showed that average temperature was a stronger predictor of *T. gondii* prevalence than maximum or minimum temperature; therefore, we retained average temperature for all analyses.*

R2, Comment 19: L160. Why use HFI over other types of measures, such as land cover, nighttime lights, pop density? I have some idea, but I also find that HFI can have some problems with it as you are lumping a lot of ideas into a single index. Need to justify.

Response: This is a great suggestion. We had selected HFI initially because we felt that it encompasses landscape change in both urban and rural environments, however, we followed this suggestion, which improved this paper. Specifically, we elected to test specific pressures individually and selected human density, crop pressure and the combined HFI. This new analysis allowed us first to test how the general association with more human-impacted environments (i.e. measured by HFI) affects *T. gondii* prevalence but then also to more specifically examine whether the association with agricultural versus urban environments presents a similar infection risk. We presented these new analyses in this paper, and they show that it is the association with urban and not agricultural environments that drives the high prevalence.

R2, Comment 20: L164. Buffer around what? Why 50 km? Also, need to describe the pixel size or minimum mapping unit of your GIS data as well as what datum and projections are being used.

Response: We have expanded on this to clarify our decision. Line 188-189 now reads: “...for each sampling location averaged over a 50 km buffer around the point.”

We discussed the rationale for the buffer width on line 189-191 which reads: “*We selected this buffer size to match the scale of geographical precision that was used as our threshold for study inclusion.*”

Also, on Line 177 we comment on the datum and projections: “*...both were projected at WGS84.*”

R2, Comment 21:L202. What are ecology variables?

Response: This sentence was rewritten and now reads on lines 241-243:

For both the ecological traits-only and climatic factor-only models, only habitat and temperature were significant and their inclusion led to increased support over the base model ($\Delta DIC=8.42$).

R2, Comment 22: L211. Move to Discussion.

Response: This sentence was removed

R2, Comment 23: L224. Please begin Discussion with answers to your predictions.

Response: We have restructured our discussion with a header sentence that summarizes our results and then expand upon these results in subsequent paragraphs. Lines 260-266 now read:

*Our global analysis indicates that *T. gondii* prevalence is associated with anthropogenic pressure, temperature and habitat across a diversity of mammalian taxa and a broad geographic range. The most important result was the strong positive association between human density and HFI with the prevalence of *T. gondii* infection in wildlife globally. In contrast, there was no support for higher prevalence of *T. gondii* in wildlife in more expansive agricultural environments as indicated by the lack of a relationship with crop coverage.*

R2, Comment 24: L226. I would disagree with this conclusion. Your work is entirely corollary and not mechanistic. A large missing component of your research is the number or density of cats in the environment or on the landscape. Essentially you are equating HFI with cat numbers, which may not always be the case. Thus, you may have found support for a prediction, but you have not really identified the mechanism here. I would certainly not call it anthropogenic spillover.

Response: We have changed the wording and have been careful avoided too strongly inferring a mechanistic explanation with domestic cats. As we have altered our analyses based on your initial recommendation, we are now looking at a separate component of HFI – human density, which does have a clearer relationship with cat density. Nevertheless, we appreciate the need to state this cautiously, given that cat density per se was not a variable in our analyses.

R2, Comment 25: L243. This paragraph is very speculative. Some of these land cover types would also simply not have cats, so is it conversion versus fact that cats are less abundant in wetlands? In the end this paragraph comes across as more of a post hoc rationalization of the analyses as you did not look at land cover/use change or other variables discussed here. This is the tradeoff of using HFI.

Response: We believe that our reanalysis of the data using human density and crop density help address Reviewer two's concerns regarding land coverage changes. Our results show that human density has strong support for increased *T. gondii* prevalence, while another measure of land use change, crop pressure completely lacked support. This provides evidence that the prevalence patterns of *T. gondii* are associated with factors that concurrently increase with human density and not simply land-use changes.

Also, in this paragraph on lines 291-297, we were summarizing an existing study that looked at how wetland degradation could lead to increased land-sea transfer of oocysts (Shapiro 2010) and a second paper that examined how impervious soil matrices influence pathogen runoff, which we discuss to put our results in the context of existing knowledge.

R2, Comment 26: L294. But how did those studies measure trophic levels? One problem with trait data is that different researchers use different values and not all of these studies may be comparable.

Response: EltonTraits is a widely utilized resource for including species-level summary diet in analyses, with over several hundred citations. We used these percentages in a qualitative manner to establish our trophic levels in very simple terms – carnivore, herbivore, invertivore and omnivore so we could correlate trophic class with *T. gondii* prevalence. We were explicit in how we defined our classifications and felt it would be more rigorous to justify our classifications on such a familiar resource. We felt that any imprecision or inherent variation associated with using such a large database would not invalidate our classification system since it was coarse and qualitative.

R2, Comment 27: L321. Again, I would disagree with this point. You essentially have a large database that is more akin to biogeography research in which you can test patterns along a descriptive nature that support or refute hypotheses. But you cannot determine mechanisms here. This is a large misunderstanding of quite a bit of the spillover literature in terms of how landscape ecology, biogeography, and macroecology evaluate the relationships.

Response: Thank you, we have revised the manuscript to remove any reference to a mechanism. We have stated that clearly in our discussion in lines 369-374 and discuss molecular epidemiologic methods that more clearly demonstrate transmission between domestic and wild cycles.

Lines 369-374 now read:

*Second, while our model can help predict global trends in *T. gondii* associated with climate and human densities, we acknowledge that identification of cause and effect mechanisms, such as domestic cat abundance, can only be deduced in a general manner at a global spatial scale. Molecular epidemiological methods are well suited to address these knowledge gaps with sensitive assays for detecting *T. gondii* DNA in water, soil or tissue samples.*

While we appreciate a strict definition of spillover (transmission to a novel host), we also intended for our definition to accompany cases when there is transmission of novel strains from the domestic cycle into the sylvatic cycle.

R2, Comment 28: L328. Agree with all of this, but wasn't really the point of the ms.

Response: We felt that a brief discussion on how this research contributes to the objective of reducing wildlife disease was warranted, so we have expanded slightly in the revision.

Lines 387-393 now read:

Preventing disease transmission between wildlife and domestic animals (and the associated spillover to humans) requires that interactions between wildlife and domestic species be restricted [2]. However, free-roaming cats kill billions of wild animals each year [65,66] and are highly invasive within most ecosystems around the world. Therefore, the intermingling of domestic cats with wildlife acts as a problematic conduit for disease transmission across the wildlife-domestic animal-human interfaces and should be limited.

R2, Comment 29: Table 1. Did you begin with an explicit framework of a priori models to consider that were based on your predictions? If so, please list these models somewhere.

Response: In our revised analysis, we slightly changed our model selection approach based on the recommendation of reviewer 1. We have elaborated on our model selection process, and lines 201-219 now read:

*We took a hypothesis testing approach and evaluated the relative support for *T. gondii* prevalence being influenced by ecological, climatic and anthropogenic factors. Model support was evaluated based on the deviance information criterion (DIC). Fixed-effect predictors within the most supported model were considered to be significant if the 95% credible interval (CI) was non-overlapping with zero. Model selection started with an intercept-only model that was used to evaluate support for a mixed-effects base model with study, species and phylogenetic effects*

*as random variables, and detection method (serology vs. isolation) as a fixed effect. This base model had substantial support over the intercept-only model, so it was used in subsequent model testing. We first assessed an ecological traits-only model, evaluating the influence of habitat or diet, separately or combined on *T. gondii* prevalence. Similarly, a climatic factor-only model was tested and the relative influence of annual average precipitation and temperature were estimated. For both of these model comparisons, the statistically significant predictors were then included in models testing the influence of anthropogenic effects. Using the base model and including influential ecological and climatic variables, we compared model support and mean effects for the three anthropogenic variables, human population density, crop coverage or human footprint index (Table 1)*

R2, Comment 30: Figure 1. One item that is apparent from this figure is that there may be some notable bias in locations sampled. There are few areas that have a low HFI and also large numbers of biomes missing.

Response: There is a concentration of research effort in particular geographic regions, but human density pressures, crop pressures and human footprint are still well sampled within those regions (e.g. a gradient from zero footprint to very high footprint). This is now discussed in lines 362-369, which now read:

*We acknowledge two important caveats in our study. First, although a large amount of *T. gondii* prevalence data was available, there are key areas such as central Eurasia and Africa that had minimal representation, which is unfortunate given that countries on these continents have relatively high human *T. gondii* prevalence [12]. Therefore, the associations found in the present analysis may not be as influential for populations in those data deficient regions. Increasing surveillance of wildlife for underrepresented taxa and regions would enable a more refined analysis for ecological and geographical variables that may influence *T. gondii* prevalence.*

As we noted earlier, we do acknowledge that there are some geographic regions lacking data and that further studies would be needed to test if the patterns we observed are consistent in those regions as well.

Appendix B

THE UNIVERSITY OF BRITISH COLUMBIA

Department of Forest and Conservation Sciences

3041-2424 Main Mall
Vancouver, BC V6T 1Z4
Tel: (604) 822-9695
Fax: (604) 822-9102

July 25, 2021

Dear Dr. Heesterbeek,

Re: Revision following the appeal of RSPB-2021-0414

We very much appreciate the opportunity to submit a second revision of this paper following the rejection of the first revision. Throughout the review process, we have placed considerable effort into addressing all editorial and reviewer concerns.

To briefly revisit the basis for our initial appeal:

Our first concern was that we felt some reviewer comments regarding the revised manuscript were incongruent with our revisions, and we received no comment from the second reviewer. We also want to emphasize here that we did conduct a complete reanalysis in the first revision. However, in this second revision, we have taken additional steps to address the reviewer's concerns and now include independent geographic data on human density and agricultural land cover. This change addresses the reviewer's primary concern on the use of the human footprint index alone as a sole measure of anthropogenic pressure and dependence on a single database. The overall results and conclusions are unchanged despite a second reanalysis, demonstrating the robustness of our results and our sincere attempt to reconcile all reviewer concerns.

Secondly, the reviewer criticisms relating to database sampling requires specific comment as to the actual statistical implications of varying sample sizes, which is not bias. Our analyses include representative gradients to test our hypotheses while controlling other influential factors, meaning that there is no location bias in our data, as the reviewer often indicates. The suggestion of bias was a statistical misinterpretation of the issue on the part of the reviewer. We do acknowledge a caveat where there are some global regions without *T. gondii* sampling, and we note how further research would be needed to confirm these patterns in those locations. Nevertheless, our study's geographic detail and representation exceed most other published macroecological studies with similar types of objectives.

Thirdly, and importantly, we adopted a global-scale synthesis, testing specific hypotheses that require parsimonious statistical modelling approaches using accessible, defensible variables. Several of the reviewer's concerns consisted of speculations of how other data that were either unavailable or spatially or temporally idiosyncratic might influence our results. Even if it were possible or appropriate to include fine-scale variables, their inclusion would lead to interdependence and model over-parameterization. It would also be inappropriate to include overly precise predictors since *T. gondii* is a life-long infection; therefore, the exact conditions at infection are always unknown. Since unexplained variance in the model is not falsely attributed to explanatory variables and our conclusions were unchanged across different landcover and human density measures, we assert that our conclusions are robust.

Periodic syntheses of emerging data are essential for building scientific consensus, reorienting or solidifying science priorities. As the reviewer pointed out, we have compiled a comprehensive dataset for a single pathogen, which supported a novel analysis. Unlike other syntheses of wildlife pathogens, our study is not limited to amalgamated pathogen guilds (bacteria vs. viruses vs. protozoans), pooled country-level data, and we were able to test prevalence and not simply pathogen richness. This level of specificity enabled us to detect a robust association between anthropogenic pressures and prevalence that is impactful and entirely consistent with the epidemiology of *T. gondii*. Demonstrating an association between anthropogenic pressure and wildlife disease contributes to the growing appreciation of the interconnectedness between wildlife, ecosystem integrity and human health. Our research approach and conclusions fit well with other recent *PRSB* papers, which also used global syntheses to address correlates of zoonotic disease prevalence (Guth et al., 2019, Johnson et al. 2020, Wardeh et al. 2020).

We request that the Editor reconsider acceptance of this paper for publication, given that we have addressed every single reviewer and editorial concern at every stage of the review process (i.e., including multiple rounds of significant reanalyses and inclusion of new data). In the context of the plethora of global-scale disease ecology research published to date, we strongly feel that a decision to reject this paper based on objections to the global variables and parsimonious analyses utilized in our study is undeserved.

We sincerely appreciated the reviewer and editorial time on this paper and thank you again for reconsidering this manuscript.

Sincerely,

Amy Wilson and coauthors

References

1. Guth S, Visher E, Boots M, Brook CE. 2019 Host phylogenetic distance drives trends in virus virulence and transmissibility across the animal-human interface. *Philos. Trans. R. Soc. B Biol. Sci.* **374**, 20190296. (doi:10.1098/rstb.2019.0296)
2. Johnson CK, Hitchens PL, Pandit PS, Rushmore J, Evans TS, Young CCW, Doyle MM. 2020 Global shifts in mammalian population trends reveal key predictors of virus spillover risk. *Proc. R. Soc. B Biol. Sci.* **287**, 20192736. (doi:10.1098/rspb.2019.2736)
3. Wardeh M, Sharkey KJ, Baylis M. 2020 Integration of shared-pathogen networks and machine learning reveals the key aspects of zoonoses and predicts mammalian reservoirs. *Proc. R. Soc. B* **287**. (doi:10.1098/RSPB.2019.2882)

Below are our more specific responses to reviewer comments/critiques:

Reviewer 1: Overall the manuscript has been greatly improved and I enjoyed reading it. Much of the revisions nicely addressed my previous comments.

Response 1: Thank you. We tried to improve the paper based on your recommendations and also integrated the ones provided in the second review.

Reviewer 2: However, I have two larger issues that remain. First, while I fully understand the appeal of the HFI, I believe that using this as your main source of environmental data is limiting your ability to get at a more nuanced understanding of the relationships you are seeking to test.

Response 2: We were a little uncertain on this suggestion, as we had indeed adopted the reviewer's first suggestion to include additional measures beyond the HFI in the first revision and so had used the underlying measure of crop pressure and density. That was a very helpful suggestion and strengthened the revised manuscript. However, to provide an even more comprehensive analysis in the second revision, we decided to use a separate measure of agricultural land cover (as the reviewer suggested) and a different measure of human population density. Thus, our measures of anthropogenic impact include crop cover, human density and the human footprint index as a composite measure. On lines 187-225 we described these databases and the variables used in our analysis. Our results with these new indices are all consistent with the results in the first revision, but the relationships are even more conclusive, and we thank the reviewer for the excellent suggestion.

Also, please note that we chose human population density rather than urban land cover because population density is a more immediate measure of the hypotheses we are testing (i.e. human activity which is expected to be correlated with domestic cats). Urban land cover in contrast, can remain even if there is no longer any human presence.

Our revision still includes climate variables from the WorldClim database to examine the relationship between temperature and precipitation and ecological and taxonomic variables. We are confident that the suite of variables selected for this analysis represents plausible hypotheses for variation in *T. gondii* across gradients of human pressure, climate and ecology. Our results with the new variables show that *T. gondii* in wildlife increase with human density but not crop pressure suggesting an association with the urban rather than agricultural environment.

Reviewer 3: I would strongly encourage the use of different data to test your hypotheses.

Response 3: In this second revision, we do use different data sets. As we described above, our variables include agricultural impact (cropland coverage), human density, the human footprint index, temperature, precipitation, ecological variables (diet, terrestrial vs aquatic environment), phylogenetic relatedness, method of detection (serology or isolation) and finally, study and species as random variables to incorporate additional unexplained variance associated with each.

Reviewer 4: The second item is in relation to the distribution of your samples. You have a wonderful database, but it is important to consider where those were collected from and why.

Response 4: We compiled data from a large number of studies (202) seeking to determine and understand the prevalence and disease cycle of *T. gondii*. We were clear in the methods that we included studies meeting certain criteria such as: only including free-ranging naturally infected wild animals and studies that included enough detailed sampling information so that we could use the data to address our research question (e.g. individual-level counts, negative and positive data and sufficiently detailed sampling location information).

With respect to the reviewer's comment on sampling location: generally the studies were associated with one taxon in one location such that when compiled in this type of macroecological study, it allows us to examine drivers of prevalence in a much broader manner. Our study does not represent a uniform sampling of the global populations, but no study could, and we have no reason to believe that there is any bias in the sampling across so many studies. Of course there are gradients in many variables across studies, but these were represented in our hypotheses and other nuisance variables we controlled for, and we are confident that there were no other variables that were excluded and would lead to a directional bias in the results. Many studies have harnessed global data to address important large-scale questions, and all of these studies, for many reasons, are to some extent geographically constrained (often much more so than our study). We have included several examples of such studies below.

Regarding the reviewer comment related to the intention of the various study authors collecting the data; researchers collecting disease prevalence data from free-ranging wildlife would be doing so to obtain estimates of local-levels of disease prevalence. *Toxoplasma gondii* is a generalist pathogen and can infect any animal, such that any prevalence data from a free-living animal that was naturally infected provides information on the exposure risk in that environment. Had we included data on experimental infections or from captive populations, we would perhaps understand the reviewer comment, but as since we did not include that type of data, we respectfully disagree with the reviewer on this point.

Selected papers using data compiled across different studies from locations worldwide

1. Gibb R, Redding DW, Chin KQ, Donnelly CA, Blackburn TM, Newbold T, Jones KE. 2020 Zoonotic host diversity increases in human-dominated ecosystems. *Nature* **584**, 398–402. (doi:10.1038/s41586-020-2562-8)
2. Murray MH, Sánchez CA, Becker DJ, Byers KA, Worsley-Tonks K EL, Craft ME. 2019 City sicker? A meta-analysis of wildlife health and urbanization. *Front. Ecol. Environ.* **17**, 575–583. (doi:10.1002/fee.2126)
3. Dunn RR, Davies J, Harris NC, Gavin MC. 2010. Global drivers of human pathogen richness and prevalence. *Proc. R. Soc. B Biol. Sci.* **277**, 2587-2985. (doi: 10.1098/rspb.2010.0340)
4. Johnson SK, Fitzma MA, Lerner DA, Calhoun DM, Beldon MA, Chan ET, Johnson PTJ. 2018. Risky business: linking *Toxoplasma gondii* infection and entrepreneurship behaviours across individuals and countries. *Proc. R. Soc. B Biol. Sci.* 285, 20180822. (doi: 10.1098/rspb.2018.0822)

Reviewer 5: I am concerned that the spatial distribution of samples is biased towards certain types of environments or locations rather than where wildlife live or are abundant.

Response 5: All global databases have some degree of disparate sampling, but we emphasize that the association that we found between *T. gondii* and human density could not be explained by the sampling distribution alone. As we described earlier, our samples are from a diverse range of free-ranging wildlife sampled across climatic and anthropogenic gradients. It is the associations with those gradients that are forming the basis of our hypothesis testing. Our dataset includes the counts of positive and negative individuals, not a single estimate of prevalence, which we have now stated in the method section:

“ From each study, we extracted data on the number of positive and negative animals...”

Therefore the underlying densities of wildlife are not influential for our analyses, and the individuals that were measured represent a sample of the population. Species vary greatly in their population densities across a range of natural and anthropogenic environments, so there would be no average location where wildlife live.

Reviewer 6: This is not a criticism of the research goals, but it is important to consider that you may have disproportionate sampling in some ecosystems or environments than others, simply based on why the original studies were conducted and these need to be considered.

Response 6: Our study did include sampling gaps as any global synthesis would, and we have acknowledged this as one of the caveats in our manuscript. Future study is needed to confirm whether the patterns we observed are consistent in those locations. The authors of all the studies used in our analysis were looking to describe local levels of *T. gondii* prevalence, and the integration of many studies across a broad gradient of

the variables of interest provide sufficient statistical power to test the association between these patterns and certain geographical, anthropogenic or climatic variables. Macroecological studies like ours nearly always use syntheses of finer-scale studies or citizen science because no study of that vast a spatial scope could be feasibly conducted. However, there is enormous value in periodically synthesizing the patterns emerging from these growing datasets. These emerging patterns advance our understanding of large-scale drivers and can be used to further develop more specific hypothesis at a finer scale. In fact, it is often global analyses that glean the most attention for advancing research agendas and action orientation by policy-based organizations.

Based on the reviewer's first review, we adjusted the text in several places (lines 271 - 273, 301 - 303) and discuss this as caveat (lines 414-422) to reiterate to the reader that there were data-deficient areas. Irrespective of poor representation in Africa or Asia for example, the global assessment conducted in this paper sheds light on driver importance under the constrained geographical disposition of the dependent variables. An excellent recent example of the use of syntheses in wildlife ecology is that of Di Marco et al. (2018) and Hill et al. (2020), which determined that mammal extinction risk and mortality (a much more complicated parameter than *T. gondii* prevalence), respectively was influenced by HFI.

1. Hill, J.E., DeVault, T.L., Wang, G. and Belant, J.L., 2020. Anthropogenic mortality in mammals increases with the human footprint. *Frontiers in Ecology and the Environment*, 18(1), pp.13-18
2. Di Marco, M., Venter, O., Possingham, H.P. and Watson, J.E., 2018. Changes in human footprint drive changes in species extinction risk. *Nature communications*, 9(1), 1-9.

Specific Comments

Reviewer 7. Suggest that title be changed to remove 'influences' and replace with 'correlates' or 'relates to' as the research is correlational, not mechanistic.

Response 7. Title now reads: "Human density is associated with the increased prevalence of a generalist zoonotic parasite in mammalian wildlife."

Reviewer 8. L15. Please thank the anonymous reviewers.

Response 8. We have thanked the reviewers in this revision and just wanted to mention we did not mean any disregard for the reviewers' time, we were adopting the custom of thanking reviewers after acceptance.

Reviewer 9. L27. Change habitat to ecosystem as habitat is a species specific term. Change throughout ms as needed.

Response 9. Thank you, ecosystem is a better term and we have changed throughout.

Reviewer 10. L35. Change ‘influence’ to ‘related to’ or ‘correlate with.’ Please revise all statements like this throughout ms.

Response 10. When discussing our models, we have changed throughout as requested, most often using “related to” , “associated with” or “relationship” depending on the context.

Reviewer 11: L53. Change agriculture to livestock as these are different. Also, while some areas (e.g., Amazonia) are expanding for livestock, in many developed nations the issues is not extensification, but intensification, such as increased densities of pigs, chickens, and cows in smaller physical spaces. Also, the point here isn’t extensification specifically, but density of the animals on that landscape. Please revise.

Response 11: We have revised this sentence as follows: “Human development and associated land use changes, can further amplify transmission risks by creating large, high-density populations of domestic animals that transmit pathogens to wildlife and humans across human-domestic animal-wildlife interfaces [1].”

Reviewer 12: L57. Please remove quotes from around terms here and throughout ms. See Goldwasser 1998 Bulletin of Ecological Society of America.

Response 12: We have removed quotes around this term (it was the only remaining case of this in the manuscript).

Reviewer 13: L65. Delete ‘extreme.’

Response 13: We have deleted this and the re-phrased section of this sentence now reads “*Toxoplasma gondii* is a generalist protozoal parasite”

Reviewer 14: L68. Just state felids as definitive hosts here. As written this sentence suggests many domestic and wild animals, which while true, could just as easily state that felids are the only group of species that are definitive hosts.

Response 14: Clarification made. Line now reads: “Domestic cats and wild felids serve as the definitive hosts in the domestic and wild cycles, respectively.”

Reviewer 15: L78. Not always. Note that there are mammals that this does not occur. Thus, while toxo is found in nearly every animal it has been sought in, there are species that simply pass it through and it doesn’t encyst.

Response 15: From our extensive literature review, we have not come across any evidence suggesting that there are any endotherms that are generally resistant to infection when challenged with viable oocysts or bradyzoites. Although many taxonomic groups have not be subject to experimental infection, so this situation remains possible. However, to accommodate the reviewer’s comment on this uncertainty, we have changed the sentence so that it now reads “*T. gondii* is a generalist protozoal parasite that is thought to be able to infect any endothermic animal...”. If the reviewer knows of studies confirming that some endotherms are immune to infection then we can revise further to acknowledge that exception with a citation.

Reviewer 16: L84. Change ‘wildlife groups’ to ‘taxa.’

Response 16: Done

Reviewer 17: L98 & 102. Both sentences begin with same wording, please revise.

Response 17: We have changed the first sentence so that it now starts with “Given these...”

Reviewer 18: L104. Indicate you mean individuals here, not species.

Response 18: We have revised so that it now reads “...with data now available for tens of thousands of individuals from hundreds of wild animal species worldwide”.

Reviewer 19: L111. Replace ‘evaluate’ with ‘test.’

Response 19: Done

Reviewer 20: L112. So is this one of your hypotheses? From here onward the ending of the Introduction is very confusing. I would suggest you combine all hypotheses and their respective predictions into a single concluding paragraph and make sure that indicate what you are hypothesizing or describe it numerically. For instance “We sought to test three hypotheses that relate to *T. gondii*. First...Second...”

Response 20: We have revised the paragraph as suggested and believe it should be clearer now. Specifically, we introduce each of the hypotheses sequentially and in the order that we tested the additive models (ecological, climatic and anthropogenic).

Reviewer 21: L112-126. These should be described before you talk about hypotheses.

Response 21: In the revised final paragraph, we first introduce the general hypothesis, followed by the citations for how prevalence should be influenced by that hypothesis (i.e. ecological, climatic and anthropogenic factors), followed by our predictions. This

is a common approach to cite the studies in this manner to lend support for what we expect and we believe will make it easier for readers to see why. In contrast, the information preceding this paragraph provides more general background on the topic. Following our adoption of the reviewer's suggestion in Response 20 we believe this will be clearer as to why we have left this material here.

Reviewer 22: L126. What hypothesis does this prediction relate to? Predictions should follow on a hypothesis.

Response 22: We have revised the paragraph now so that for each of the main potential drivers (e.g. anthropogenic) we first state what it is, followed by how it might influence *T. gondii* prevalence in wildlife using other studies (as described in comment 21) followed by our prediction. We believe this section is now clearer following the reviewer's suggestion on how to better structure it.

Reviewer 23: L129. Is this the second or third hypothesis?

Response 23: The second relates to climatic factors and the third to anthropogenic factors. This should be clearer now as described in our previous comments.

Reviewer 24: L131-5. These would read more clearly if you state the prediction and then follow it by the rationale. As written they are awkwardly.

Response 24: As described above, we have stated the general hypothesis, followed by how it is thought to be impactful, followed by the prediction. We believe this whole section is clearer now.

Reviewer 25: L142. Should also state if you followed the PRISMA procedure for conducting reviews. This should be provided as a supplement showing the numbers at each stage of process.

Response 25: We did not follow the PRISMA procedure formally *a priori* but had provided some breakdown of papers included or excluded in the supplementary information table S1 and figure S2.

Reviewer 26: L152. Ok, but I would argue this is still a weak justification. Did you go through and calculate dispersal/movement based on an allometric equation by species? Given that you are dealing with such a wide variety of taxa your dispersal data are going to vary markedly. Thus, rather than an arm-waiving description here you do need to calculate something like median and mean dispersal and perhaps group it by major taxa, and then demarcate your boundary.

Response 26: To address the reviewer's concern regarding dispersal distance, we have removed the reference to species distance, which we only meant in a very general

way. This was not arm-waving but instead reflects an appreciation of the variability of dispersal. Through our own studies of marked individuals and the consensus in the literature, dispersal ranges of species are highly context-dependent and change with individual age, gender and landscape configuration within the population. Moreover, dispersal estimates are only available for approximately half of the 232 species included in this study through the PanTHERIA database (Jones et al. 2009). Thus, it would not be possible to identify a separate mean dispersal distance in this case and use that to define the radii differently for each species.

We want to reiterate that global macroecological syntheses like this require certain types of decisions for feasibility, and this includes a decision on a focal radius within which variable information is derived. This is very standard even in finer scale landscape ecology studies. We also feel that it is important not to suggest certainty and precision that is not present, which we made a concerted effort not to do in this manuscript. Our decision to adopt a 50km buffer was chosen to more accurately represent the precision of the location information from each study and to reflect the likely geographical and climatic factors experienced by the individuals. The additional variability from differences in dispersal distances across species (or different age and sex classes within species) is not a source of directional bias but instead contributes to additional unaccounted for error variance. There will be many such sources of unaccounted for variance as there are in all ecological studies. In this revision, we have stated this:

“For our analyses, we selected a 50 km threshold in order to reflect the average precision of sampling information provided by authors and to more realistically represent the geographical gradients experienced by individuals”.

Reviewer 27: I know this is an extra step, but it is important that you have a fully justifiable choice for selecting a bound that is expected in landscape ecology and biogeography.

Response 27: Thank you for this perspective. As we discussed in response 26, selecting a defensible boundary is an expectation in landscape ecology and biogeography but another expectation is that we do not misrepresent the precision of our estimates, which we did by selecting a 50 km buffer. Decisions like this are always present in macroecological studies that use information from different sources. It would be unreasonable to come up with a justifiable species-specific buffer, as we described more fully in Response 26. Most species in this study lack movement information, so this requirement would necessitate limiting our analysis to well-studied species, further contributing to reduced sampling coverage.

Reviewer 28: L155. How are anthropogenic pressures consistent over a 50 km boundary? What are you measuring as anthropogenic here, land cover, air pollution, etc.?

Response 28:

We did not intend to say that anthropogenic pressures were always consistent over a 50km boundary; that was only the case in some very remote locations and so we have removed that sentence.

Reviewer 29: L168. Please indicate what Elton trait groups were merged into your foraging groups or give specific criteria on how percentages resulted in a species being defined as one of the four groups in your research. Essentially you need to tell the reader how if they went to Elton traits that their classification gets transitioned into yours so they could replicate.

Response 29: We had provided percentages for omnivores and carnivores and have now specifically included the 50% threshold for herbivores and invertivores. We have also have provided a supplementary table S2 for readers wanting to see a more detailed classification. The lines now read:

“We reclassified these proportions into four categories: herbivores, invertivores, omnivores and carnivores. Herbivores and invertivores were species where > 50% of the diet was vegetation and invertebrate prey, respectively. We defined omnivores as species where the primary diet could be vegetation or invertebrates but included <50% vertebrates. Carnivores were species that primarily consumed >50% live vertebrate prey or carrion. Species-specific dietary compositions and resulting classifications are provided in supplementary table S2.”

Reviewer 30: L180. Comma after maximum.

Response 30: We corrected this.

Reviewer 31: L183. This gets back to my first review. HFI is a very mixed bag type of metric as it simply provides some score based on these other various inputs without expressly separating each. My problem, as I mention previously, is that the HFI does not include a lot of other measures of human influence, and is a very generic measure that lacks the kind of information that you are seeking to try to evaluate in your models.

Response 31: Please see our comments 2 and 3 but to reiterate, we no longer focus only on the HFI and instead include two specific measures, cropland coverage and human density, to reflect agricultural and urbanization impacts, respectively. We still use the HFI as a third measure, and compared with the two specific measures (human density and cropland), HFI is useful as a comprehensive measure of anthropogenic pressure. However, we felt that the reviewer’s original recommendation to think beyond HFI alone was excellent, and have followed up on that recommendation.

Reviewer 32: My concern is that this is an easy to access database that provides a single gross level metric that misses the nuances or background rationales for evaluating specific environmental attributes.

Response 32: As we discuss in responses 2, 3 and 31 for both revisions, we did not only focus on the HFI. In the first revision, we used HFI and crop pressure and human density pressure, all from Venter et al. (2016). In this revision, we used HFI and then crop coverage and human density from independent databases. We retained HFI as one of three anthropogenic variables because we consider the HFI to be a useful, composite variable that captures quantifiable anthropogenic pressures. To reiterate, HFI includes eight variables: 1) extent of built environments extent, (2) cropland, (3) pasture land, (4) human population density, (5) night-time light (6) railways, (7) roads and (8) navigable waterways that are combined into a single index. By including all of these pressures, the HFI will be highly correlated with other less quantifiable measures such as air and water pollution, noise disturbance, waterway disruption, invasive species infiltration and pesticide exposure, all of which would influence wildlife health. The HFI database (Venter et al. 2016) has over 600 citations and has been used in other global syntheses as an index of anthropogenic pressure. The widespread scientific usage, accessibility and defensible use of metrics in the HFI database is a tremendous benefit and was another reason for us selecting it for our original manuscript and for retaining it in the revisions. Nevertheless, we appreciate the reviewer's suggestion to use independent databases for agriculture and human density and doing so has strengthened our manuscript.

Reviewer 33: While the database has value, I remain unconvinced here that this was the best one to use really go into depth on factors related to wildlife and disease. I would suggest redoing the analysis with land cover data and other separate databases that can give more support for specific relationships.

Response 33: We have addressed this concern as described earlier; please see Responses 2, 3, 31 and 32.

Reviewer 34: L211. I would suggest you specifically note which models or model sets are being used to test each hypothesis.

Response 34: We have more carefully detailed the hypotheses in the introduction and also have provided additional text in table 1 reminding readers of the classifications and hope that the facets and symbols in figure 2 provide additional guidance.

Reviewer 35: L234. This sentence doesn't make sense as written.

Response 35: This line now reads: "Our literature search produced 202 *T. gondii* prevalence studies representing 238 species from 54 taxonomic families and 45,053 tested individuals (electronic supplementary, table S1)."

Reviewer 36: L241. Comma after species.

Response 36: We made this correction.

Reviewer 37: L245-9. Here it sounds like you broke apart the HFI, but that is not how you described it in Methods.

Response 37: In the previous version, we had used elements from the HFI, and we realize we were not clear enough in how that was done. However, in this present version, we have used independent variables and our use of those is described in the Methods text.

Reviewer 38: Reviewer: L265. But this should be completely expected as would not expect a lot of wildlife in strict agricultural lands and also I would expect that sampling effort for toxo is low in these areas. This brings up an important point of sample bias across ecosystems. If we use land cover a proxy for ecosystem, where are most of samples from?

Response 38:

This sentence was removed in this revision. However, we want to address this point. First, *ecosystem* in this paper refers to terrestrial or aquatic which is the term that you requested we use instead of habitat. Many species of wildlife choose or are forced to live in and adjacent to the boundaries of agricultural lands. In contrast to the reviewer's assumptions, studies frequently occurred in agricultural areas due to the concern for rodent or avian wildlife acting as intermediate hosts (and wildlife diversity and abundance can often be high in agricultural landscapes although that declines with extreme extensification and intensification). In addition, it is important to point out that *T. gondii* prevalence within sampled animals is not dependent on sampled animal density as animals can be infected from an environmental stage even in areas with low density, though as above we do not expect wildlife densities to necessarily be low in agricultural regions. We have ensured that readers understand that we implemented count data which now read: "Prevalence data was modelled as counts and assumed a multinomial distribution."

Again, we disagree that varying sample size constitutes bias. It is the categorization of those samples into categories that could introduce statistical bias. For example, if we utilized only carnivores in the terrestrial grouping and only herbivores in the aquatic group (which we did not do) without controlling for diet, then we would falsely assume that terrestrial habitat had a higher risk. We are avoiding bias by using defensible categories and we are using quantitative metrics to test for associations between geographical variables.

Reviewer 39: Forested locations, grasslands, urban areas? This is actually an issue that needs to be considered as the lack of relationships may simply be due to where the animals were sampled in the first place and hence strong inequities occur. Certainly for wild birds and mammals in the US, where we sample for diseases varies markedly.

Response 39: We want to reiterate that the sampling locations used in this analysis included broad gradients for the ecological, climatic and anthropogenic variables we tested in this study, and this is what is needed for the statistical tests of our hypotheses. By using landscape measures such as crop pressure, human density and human footprint, we are also accounting for these changes. For example, the correlation between agricultural or urban land cover and forest cover is often in the $r = -0.80$ to -1.0 range. In this manuscript, we adopted a hypothesis-driven approach and tested particular landscapes with variation in the ecological, climatic and anthropogenic gradients for our hypotheses. We then interpret a lack of a relationship between cropland coverage as being suggestive that human density and other factors that we have not included in our model are more influential than cropland coverage as defined. We respectfully disagree with the reviewer's suggestion that geographic sampling alone has created this association. We reiterate again that since we are dealing with raw count data, different sample sizes and the associated sampling error are dealt with in the statistical model.

We agree with the reviewer that diseases in wildlife are not surveyed at random, and often data is obtained opportunistically. However, we strongly disagree that this sampling negates the use of this data in large macroecological analyses. Most citizen science data for example, is also obtained opportunistically and yet it has become an extremely valuable source of ecological data. The important thing is that the samples are represented from across gradients of the variables of interest and that other sources of variance are accommodated to the extent possible. On this latter point, please also note that by including study as a random variable, we are accounting for study specific differences that might further contribute to variance in the results.

Reviewer 40: L301. Yes, but how are your wildlife samples distributed across space? Are people sampling wildlife in any large numbers near farms? I think the problem here is several fold. Barn cats are localized and densities may be low, especially with increasing distances from farm. Also, what wild animals are on farms or near them? Many wildlife do occur in ag landscapes, but many do not.

Third, seasonal mismatch. Even if wildlife are present are they in these locations at a time when they might interact with cats? Finally, sampling effort in these landscapes. How representative of a sample do you have in agricultural areas?

All of these points lead me back to earlier comments about need to describe the spatial distribution of your samples across land cover or ecosystem types (if we don't have enough samples from some contexts, can we really test for relationships?) and also the lack of separate measures of the environment (rather than just HFI).

Response 40: We have adjusted this line to reflect the rural cat density data. Lines now read:

“Due to the significant landscape and hydrological alterations associated with agricultural intensification, we had predicted that *T. gondii* prevalence in wildlife would be positively associated with cropland cover. However, we did not find a consistent association with cropland coverage. Native habitat retention within agricultural landscapes has been associated with lower risks of pathogen transfer from domestic animals into wildlife [59]. However, domestic cat densities frequently show a reduced [60], but unequal distribution across rural habitat types [53,61], and this may complicate attempts to elucidate a broad-scale pattern using available land cover metrics.”

In relation to the reviewer’s comments regarding seasonal mismatch; as we stated in the introduction, *T. gondii* oocysts can move through the environment passively and remain viable across seasons and years, so wildlife would only need spatial and not temporal overlap with cats in order to become infected. When these oocysts are transferred to watersheds from runoff, the chance of wildlife being exposed increases as discussed in the manuscript. The duration of viability and passive spread of oocysts also means that direct interaction with cats is not necessary, and wildlife can be infected by eating infected intermediate hosts that were in contact with cats both spatially and temporally.

The reviewer has again mentioned the issue of sampling distribution across space. We have an adequate sampling distribution across our variables of interest, and in all manuscript versions we provided frequency histograms in the supplementary information so that readers can review this distribution. The wildlife samples were collected across gradients of cropland coverage and the location information for specific farms is not accessible information at the scale at which this study was conducted, which is why the HFI rasters provided by Venter are so valuable. We also reiterate that our inclusion of study as a random variable helps account for any location-specific sources of variance that were not included in the variables used to test our main hypotheses.

We had also addressed the reviewer’s concern about using separate measures of the environment in the initial revision and have done so again in this revision by utilizing different variables for cropland and human density that do not originate from the geographical database from which the HFI is calculated.

Reviewer 41: L310-15. I would suggest cutting this part as it is very speculative. Also, we know that species diversity declines with latitude and that there are many other ecological phenomena that have latitudinal gradients. Thus, without including such evaluations in your analysis, you really can’t say more. I don’t disagree that we find more cats in warmer areas, but you don’t have data to say much aside from that you found a relationship.

Response 41: This paragraph has been completely rewritten.

Reviewer 42: L318. It isn't simply the average rainfall, but how much occurs per unit time, especially when topography is included. To move oocysts over land requires overland flow, which requires a larger volume of water in a short amount of time. All this is to say that frequency and amplitude of precipitation matter greatly.

Response 42: For a global study such as this, average rainfall is an appropriate predictor given that rainfall intensity and topography are too spatially and temporally inconsistent to be included in any macroecological study. Firstly, precipitation of any amount will cause oocyst movement through soil, but as VanWormer et al. (2016) suggest, the speed of this movement depends on having a threshold volume of precipitation and mobility differs through soil type. Perhaps most importantly, please note that even at the scale of the Californian coast, VanWormer et al. (2016) did not use empirical weather data and topography; they instead used a modelling approach – they modelled different precipitation rates with three general land cover classes (1: urban or rural, 2: agriculture, 3: undeveloped).

Thus, we felt it is important to use appropriately specific geographical variables for global analyses for determining general associations, which can, of course, be used as a catalyst for more nuanced microspatial studies where more precise predictors could be deployed with more confidence. Again we emphasize that animals are thought to be infected with *T. gondii* for life, and we use prevalence data. Therefore, it would be impossible to know the exact climatic conditions at infection, making it inappropriate to use temporally precise climatic variables.

Reviewer 43: L360. Also the strain/genotype of toxo matters. I think if your phylogenetic comparison doesn't include strain, that you are also missing a large component of possible explanatory power.

Response 43: We agree that the strains of *T. gondii* differ with respect to disease severity and infectivity, which was discussed in the text lines 324-329. However, the majority of studies do not allow for strain identification since they rely on seroprevalence, so would only indicate exposure and not strain. Unfortunately, therefore, strain information would not be available for enough samples to be included as a variable in our analysis, but could be a future area of study as the available data accumulates. We should point-out that strain identification cannot be done without sacrificing the host animal for tissue isolation, which introduces study constraints and ethical issues. We also note that although certain strains predominate in sylvatic versus domestic cycles, they can occur in both cycles (Shapiro et al. 2019). Even outside strain identification, the documented association with human density is highly suggestive that domestic cats are the most consequential source.

Reviewer 44: L363. Other possible/noted limits are (or maybe) unequal representation of samples across land cover/ecosystem type, and lack of strain/genotype information.

Response 44: We have addressed this in our earlier responses, but we reiterate that global data cannot comprehensively include all ecosystems, climate, and geopolitical situations, however, this does not disqualify its use. As the example global studies provided below indicate, the use of compilations of global summaries of disease prevalence data can provide considerable insight and impact on knowledge and policy when presented and analyzed in a manner that doesn't overreach the data.

1. Gibb R, Redding DW, Chin KQ, Donnelly CA, Blackburn TM, Newbold T, Jones KE. 2020 Zoonotic host diversity increases in human-dominated ecosystems. *Nature* **584**, 398–402. (doi:10.1038/s41586-020-2562-8)
2. Dunn RR, Davies J, Harris NC, Gavin MC. 2010. Global drivers of human pathogen richness and prevalence. *Proc. R. Soc. B Biol. Sci.* **277**, 2587-2985. (doi: 10.1098/rspb.2010.0340)
3. Wells K, Gibson DI, Clark NJ, Ribas A, Morand S, McCallum HI. 2018 Global spread of helminth parasites at the human-domestic animal-wildlife interface. *Glob. Chang. Biol.* **24**, 3254–3265. (doi:10.1111/gcb.14064)
4. Clark NJ, Seddon JM, Šlapeta J, Wells K. 2018 Parasite spread at the domestic animal - wildlife interface: anthropogenic habitat use, phylogeny and body mass drive risk of cat and dog flea (*Ctenocephalides* spp.) infestation in wild mammals. *Parasit. Vectors* **11**, 8. (doi:10.1186/s13071-017-2564-z)
5. Civitello DJ *et al.* 2015 Biodiversity inhibits parasites: Broad evidence for the dilution effect. *Proc. Natl. Acad. Sci. U. S. A.* **112**, 8667–8671. (doi:10.1073/pnas.1506279112)

Reviewer 45: L388. These last two paragraphs, especially the last one, are not strongly related to the work. They're good pieces of information and I don't disagree with them, but they make the Discussion longer and extend beyond what your work really was centered on.

Response 45: Thank you, we have removed this paragraph and integrated the message of cat impact in the concluding paragraph.

Reviewer 46: L400. What are landscape ecosystem services?

Response 46: We had meant references to the ecosystem services provided by the landscape but this paragraph has been restructured and clarified, and the line now reads: "Mitigating these influences through landscape restoration"

Appendix C

THE UNIVERSITY OF BRITISH COLUMBIA

**Department of Forest and
Conservation Sciences**
3041-2424 Main Mall
Vancouver, BC V6T 1Z4
Tel: (604) 822-9695
Fax: (604) 822-9102

September 14, 2021

Dear Dr. Heesterbeek,

Re: Revision of RSPB-2021-0414

Thank you for offering us the opportunity to revise our manuscript further. We have made every effort to accommodate the associate editor and reviewer's recommendations. On the attached response to reviewers, we have included our response underneath the comment raised by the editor or reviewer, which has been bolded for clarity. We adopted all remaining minor comments, but we wanted to highlight several points briefly. First, the reviewer suggested repeating the analysis at different scales, which we have done by recalculating the variable values for a 25 km buffer. The model results were qualitatively and quantitatively consistent with the results obtained for a 50km buffer. Second, the reviewer brought up the issue of adjusting the buffer to reflect dispersal distances for those species where estimates exist. Due to inherent variability and context dependency of dispersal distances, we believe that adjusting our analyses would be problematic. Furthermore, the majority (78%) of individuals in the dataset have dispersal ranges much smaller than 50 km, which was also chosen to reflect the precision in sampling locations. However, we have expanded on this in the manuscript to better justify our buffer choice. We sincerely appreciate the reviewer and editorial time on this paper and thank you again for offering us the opportunity to revise our manuscript.

Sincerely,

Associate Editor

Comments to Author:

After reading this careful and extensive revision by Wilson et al., a reviewer has offered a few comments and suggestions. In preparing a response to reviewer comments, please indicate further edits that have been made to the manuscript to address these remaining suggestions or please offer explanation for when there is a difference of opinion with the reviewer. Some additional methodological details also may benefit the manuscript, including details on GIS/imagery data (projection and datum used, size of pixels or minimum mapping unit, when data are from) and details on editing/prepping the data, evaluating it for use in a GLM, and details on model building.

Response: In our revision, we have provided information on the projection, resolution and source for all GIS data used. We have also added some additional details on the preparation of the data and evaluation of model fit. With respect to fit, there is no current approach analogous to an R² statistic for phylogenetic generalized linear mixed models run with a Bayesian framework. We have used statistics to evaluate the model performance and convergence and note these in the manuscript (lines 275-281). We also refer to the strength of the coefficient effect sizes in assessing support for models with influential variables (in addition to DIC) in lines 251-254.

Additionally, we have made some adjustments to how the model building is reported and shown in Table 1, based on the suggestion from the reviewer. We agree that this is a better way to report on the models given how we tested the hypotheses rather than our previous presentation.

Reviewer(s)' Comments to Author:

Referee: 2

Comments to the Author(s).

General Comments

Overall the manuscript has been improved further from my previous evaluation of it. While improved, there remain a handful of items that need further attention.

First, in terms of the statistical analyses and model building, now that you have clarified your three hypotheses and predictions, a bit more clarification is needed. Specifically, prior to model building you need to test your independent variables for collinearity and remove variables if needed (if you did this, then let reader know).

Response: We had tested but not reported the correlations of the different predictor variables in earlier versions but now report these in a correlation matrix in the Supplementary Material Table S3. There are no mean correlations that exceed 0.7, which is often used as a criterion for when two variables should not appear in the same model. However, temperature and cropland cover are close to this cut-off at 0.69. We assessed how the other coefficients in the model vary when temperature is not included in the same model and have shown these below

in the case for the top model and the model containing crop cover. There are only very slight changes to the posterior means and credible intervals for all of the coefficients when temperature is and is not included in the same model (shown below) and therefore, we have retained it in all the models with density, footprint and crop cover. This is discussed in the manuscript in lines 226-232.

Global top model with temperature: ecosystem + temp + human density + base

	post.mean	l-95% CI	u-95% CI	pMCMC
(Intercept)	-1.69827	-3.68957	0.55311	0.1027
Ecosystem	-0.96237	-1.69195	-0.24719	0.0127*
temp	0.43366	0.22658	0.62776	<3e-04***
Human density	0.71774	0.41414	1.03355	<3e-04***

Global top model without temperature: ecosystem + human population density + base

	post.mean	l-95% CI	u-95% CI	pMCMC
(Intercept)	-1.8417	-4.0033	0.2869	0.0813
Ecosystem	-0.9326	-1.6577	-0.1831	0.0153*
Human density	0.7649	0.4503	1.0749	<3e-04***

Model including crop with temperature: ecosystem + temp + crop + base

	post.mean	l-95%CI	u-95% CI	pMCMC
(Intercept)	-2.1974	-4.16145	-0.15038	0.0327*
Ecosystem	-0.93159	-1.64563	-0.20672	0.0107*
temp	0.45743	0.26175	0.65744	<3e-04***
cropland	0.05044	-0.19576	0.30123	0.678

Model including crop without temperature: ecosystem + crop + base

	post.mean	l-95% CI	u-95% CI	pMCMC
(Intercept)	-2.35926	-4.67878	-0.4356	0.0253*
Ecosystem	-0.93735	-1.71664	-0.23321	0.012*
cropland	0.03931	-0.21945	0.33149	0.78

In addition, while I understand the linear nature of the models, no reason was provided why only linear terms were used (i.e. no quadratic terms) or interactions. These may not be needed, but it would be helpful to clarify here and if they should be considered, please do.

Response: At the beginning of this analysis, we discussed the nature of the hypothesized relationships and what response would be the best fit given the a priori predicted responses and the range of the predictor variables in our data set. Based on the objectives of these analyses, we decided that a linear fit was the best

and most reasonable description for the climate and land cover variables and therefore limited our analyses to those. We have now included a sentence on lines 249-251 to describe this. Some alternative patterns may exist other than those we tested. However, because our analysis was already complex and we based the shape of the relationship around specific hypotheses, we chose not to test too many alternative patterns with higher-order polynomial terms or interactions. That is certainly something that other authors could explore further with a simpler analysis and smaller scale.

Since you have three hypotheses, it would make more sense to have three separate model groupings that are being compared within a set. From Table 1 it appears that these are all the models you ran, rather than model 1-5 are in relation to testing hypothesis 1, models 6-10 are in relation to testing to hypothesis 2, etc. Such a mirroring of analysis relative to hypothesis should be done.

Response: We have adopted the reviewer's suggestions and have simplified the table and organized it with hypotheses instead of the previous organization. The only change from the original table is the removal of the base model details, which was already described in the text and in the table legend.

A final analytical point is that DIC interpretations are fine, but need to describe what level a model is no longer considered in top model set (similar to $AIC > 2$) and also if cannot provide some measure of fit (e.g., akin to adjusted r^2) it needs to be noted that you have top models, but that they might not describe much (this is an issue that readers need to be reminded of as AIC, DIC, etc., are valuable for informing models relative to one another, but they don't provide a measure of how much variance is described in an easy to interpret manner. This does not mean the work is problematic, rather it provides further contextualization of the research which is needed to make your findings crystal clear to readers).

Response: We now discuss the DIC model selection thresholds (lines 252-253) and refer the reader to the appropriate source (Spiegelhalter et al. 2002)

Second, while I appreciate the authors view on bias, they are not necessarily accurate. This is not a point I wish to argue with the authors about, but having published widely with monitoring data and in macroecology there are numerous ways we treat data and account for it prior to statistical analyses that the authors have not done. Statistical analyses alone simply does not account for or removes bias. The points raised about lack of geographic coverage, effort, and time of year are all ways that the data may be biased that are not very well accounted for in the work. I consider the work in this manuscript as critically valuable and my points here are not to upset you the author as much as to make sure the work is bullet proof given the many controversies that any work involving cats entails. At a minimum all of these points need to be noted in the limitations paragraph of the discussion.

Response: We thank the reviewer for acknowledging the importance of this work. We appreciate the extensive effort that the reviewer has invested in ensuring the

research was the best it could be. We have made every effort within our capacity to comply with their suggestions. Our overall point in defence of our work was that valid but cautious interpretation could be made with datasets that have adequate coverage across the variables as tested. We believe that while our analyses may have gaps in geographic coverage, variation in effort or sampling by season, we don't have reason to expect that this alone would lead to a directional bias that would affect the conclusions of our study. These are common considerations in macroecological studies based on citizen science or meta-analyses and may lead to greater variability in the responses but not bias *per se*. We note that despite the expected variability in the data due to these gaps in coverage or differences in survey effort, our results are highly constant. The strength and direction of associations between different models, spatial scales or even across predictor variables (e.g. human density or anthropogenic footprint show the same result) have remained consistent. This consistency provides support for our results. We have carefully considered the reviewer point regarding representation in different environments and have expanded on it in the manuscript in lines 433-437.

Lines 435-439: *Furthermore, we broadly categorized ecosystem types as terrestrial or aquatic, and sampling efforts are unlikely to be uniform across different environments and habitats within these ecosystems. Therefore, the associations found in the present analysis may differ from those present in wildlife populations in those data-deficient regions or environments.*

We would also like to highlight how we have controlled the influence of several important sources of uncertainty in our analyses:

1. Implementing a *quantitative* phylogenetic correction, which helps controls for the myriad of reasons as to why phylogenetically closer species may have similar prevalence (e.g. similar ecological traits, geographic ranges, dispersal movements and sensitivity to *T. gondii* which was discussed in the manuscript in lines 421-429).
2. Including a random effects variable for study, which accounts for study-specific idiosyncrasies such as researcher methodologies, location, or season.
3. Including a random non-phylogenetic effects variable for species, which accounts for ecological traits that influence prevalence patterns which are not encompassed by the ecological variables included in this analysis.

Discussion.

Third, the argument about the area used as a landscape (50km²) skirted the issue quite a bit. For instance, you noted that for ~50% of the spp you could have found body size and estimated ranges. Yes, this would cut your spp sample size down, but the work would have greater ecological relevance by evaluating the landscape elements that lie within the dispersal distance of an average mammal (you could also have this analysis as a more in-depth approach and then used all spp in a more basic approach). I really was just not convinced that 50km is an ecologically justified distance (yes, dispersal and home range vary by season, location and spp, but we do know that on average a large mammal moves farther than a small mammal), especially with the diversity of mammals evaluated. Many macroecology studies do

not simply use one scale, unless they have based it on something like similar spp or allometry. Minimally, it seems like an evaluation at several scales would have been done here or a subsetting of the data to look more closely at the spp for which you can find body mass data. The argument that other landscape ecology studies follow what you have done certainly occurs at times, but there is a strong literature pointing out that for landscape ecology to advance and be done well that it needs to have non-arbitrary bounding. The scale here is not based on animal movement, political boundary, etc.

Response:

The major rationale of our choice of a buffer was driven by the precision of the sampling locations, which is a separate consideration from the dispersal distance of the species. With the data in mind, a buffer of 50 km most realistically captures the sampling precision. However, to address the reviewer recommendations, we have tested an additional buffer range of 25 km, to examine the consistency of the results. We found results between a 25 and 50 km buffer to be concordant, thus further showing the robustness of the study's findings. We note this additional scale in the methods and include a table of the coefficients as supplementary material but otherwise focus on the 50 km scale.

We have further addressed the reviewer's recommendations to justify our buffer width in lines 167-171 (below), and we also include the home range sizes in the existing supplementary table S1.

Lines 167-171: *Of the 238 species in this study, a subset of 142 had species-specific home range estimates available [31]. The home range estimates were much smaller than 50 km for 127 of these species, with individuals from these taxa constituting 78% of the full dataset and 93% of the subset of taxa with estimated home ranges (supplementary table S1).*

While an analysis that identifies anthropogenic influence/land cover within the dispersal range of all species would be interesting, we feel that it would require a more specific and localized study for several reasons. First, intraspecific dispersal distances can be highly variable due to age, sex, habitat quality or time of year (and likely others). Therefore, adopting a single average value or an extrapolated value could be potentially misleading if used in this particular analysis. We intentionally adopted diet categorizations for that reason. Furthermore, the extent to which dispersal tendency affects infection risk depends on the spatial scale of oocyst contamination. Even highly vagile species may spend the majority of their time in very anthropized environments if the spatial scale of that disturbance is large. Second, the species for which home range has been estimated, are not representative samples of mammals. One of the strengths of this work is that it includes such a large range of taxonomically diverse species, ensuring that our patterns are robust across a range of species, not just commonly studied ones.

We also would like to reiterate that by including a quantitative phylogenetic component in this analysis, we are implicitly accounting for species-specific

characteristics that could influence prevalence and this includes dispersal patterns (or movement more generally). The phylogenetic signal in our analysis was strong and likely due to other ecological traits besides habitat and diet, which we discuss in lines 417-421. Our goal was to test specific hypotheses for how anthropogenic environments and climatic conditions influence *T. gondii* prevalence generally in mammals and so we did not test all possible ecological traits. However, more specific analyses that look at the influence of traits like movement would be best accomplished with a study focusing on fewer taxa (or single species) experiencing the same spatial scale of pathogen contamination.

Fourth, the Discussion should begin with answers to your hypotheses. Did you find support?

Response: We have reworded the text in the opening paragraph of the Discussion so that it is framed in terms of support for our hypotheses related to human density (lines 320-323) and then again for the case with climate (on lines 369-371).

Fifth, the revised ms has a variety of spelling and grammatical items that should be revised (data are rather than data is, depredated instead of predate as predate means to occur before, short paragraphs that need to be combined with others as a paragraph should have at least three sentences, etc.).

Response: Thank you for noting these, we have corrected any remaining spelling and grammatical errors.

Attending to these aspects will very much help improve the work in a manner that will advance our understanding of this parasite and its implications in a One Health context. I very much understand that my points require further work, but believe accounting for these considerations will yield a widely read and cited ms that has real world implications.